# DOMAIN ADVERSARIAL TRAINING
# A GAME PERSPECTIVE

**David Acuna**[123]**, Marc T. Law**[2]**, Guojun Zhang**[34]**, Sanja Fidler**[123]

[1]University of Toronto  [2]NVIDIA  [3]Vector Institute  [4]University of Waterloo

## ABSTRACT

The dominant line of work in domain adaptation has focused on learning invariant representations using domain-adversarial training. In this paper, we interpret this approach from a *game theoretical perspective*. Defining optimal solutions in domain-adversarial training as local Nash equilibria, we show that gradient descent in domain-adversarial training can violate the asymptotic convergence guarantees of the optimizer, oftentimes hindering the transfer performance. Our analysis leads us to replace gradient descent with high-order ODE solvers (i.e., Runge–Kutta), for which we derive asymptotic convergence guarantees. This family of optimizers is significantly more stable and allows more aggressive learning rates, leading to high performance gains when used as a drop-in replacement over standard optimizers. Our experiments show that in conjunction with state-of-the-art domain-adversarial methods, we achieve up to 3.5% improvement with less than half of training iterations. Our optimizers are easy to implement, free of additional parameters, and can be plugged into any domain-adversarial framework.

## 1 INTRODUCTION

Unsupervised domain adaptation (UDA) deals with the lack of labeled data in a target domain by transferring knowledge from a labeled source domain (i.e., a related dataset with different distribution where abundant labeled data already exists). The paramount importance of this paradigm has led to remarkable advances in the field in terms of both theory and algorithms (Ben-David et al., 2007; 2010a;b; Mansour et al., 2009). Several *state-of-the-art* algorithms tackle UDA by learning domain-invariant representations in an adversarial fashion (Shu et al., 2018; Long et al., 2018; Saito et al., 2018; Hoffman et al., 2018; Zhang et al., 2019; Acuna et al., 2021). Their goal is to fool an auxiliary classifier that operates in a representation space and aims to classify whether the datapoint belongs to either the source or the target domain. This idea, called *Domain-Adversarial Learning* (DAL), was introduced by Ganin et al. (2016) and can be more formally understood as minimizing the discrepancy between source and target domain in a representation space (Acuna et al., 2021).

Despite DAL being a dominant approach for UDA, alternative solutions have been sought as DAL is noticeably unstable and difficult to train in practice (Sener et al., 2016; Sun et al., 2019; Chang et al., 2019). One major cause of instability is the adversarial nature of the learning algorithm which results from the introduction of the *Gradient Reversal Layer* (GRL, Ganin et al., 2016) (Figure 1). GRL flips the sign of the gradient during the backward pass, which has profound implications on the training dynamics and asymptotic behavior of the learning algorithm. Indeed, GRL transforms gradient descent into a competitive gradient-based algorithm which may converge to periodic orbits and other non-trivial limiting behavior that arise for instance in chaotic systems (Mazumdar et al., 2020). Surprisingly, little attention has been paid to this fact, and specifically to the adversarial component and interaction among the three different networks in the algorithm. In particular, three fundamental questions have not been answered from an algorithmic point of view, *1) What is optimality in DAL? 2) What makes DAL difficult to train and 3) How can we mitigate this problem?*

In this work, we aim to answer these questions by interpreting the DAL framework through the lens of game theory. Specifically, we use tools developed by the game theoretical community in Başar & Olsder (1998); Letcher et al. (2019); Mazumdar et al. (2020) and draw inspiration from the existing two-player zero-sum game interpretations of Generative Adversarial Networks (GANs)

(Goodfellow et al., 2014). We emphasize that in DAL, however, we have three rather than two networks interacting with each other, with partial cooperation and competition. We propose a natural three-player game interpretation for DAL, which is not necessarily equivalent to two-player zero-sum game interpretations (see Example 1), which we coin as the Domain-Adversarial Game. We also propose to interpret and characterize optimal solutions in DAL as local Nash Equilibria (see Section 3). This characterization introduces a proper mathematical definition of algorithmic optimality for DAL. It also provides sufficient conditions for optimality that drives the algorithmic analysis.

With our proposed game perspective in mind, a simple optimization solution would be to use the Gradient Descent (GD) algorithm, which is the *de facto* solution but known to be unstable. Alternatively, we could also use other popular gradient based optimizers proposed in the context of differentiable games (e.g. Korpelevich, 1976; Mescheder et al., 2017). However, we notice that these do not outperform GD in practice (see § 6). To understand why, we analyze the asymptotic behavior of gradient-based algorithms in the proposed domain-adversarial game (§ 4). The main result of § 4.2 (Theorem 2) shows that GD with GRL (i.e., the existing solution for DAL) violates the asymptotic convergence guarantees to local NE unless an upper bound is placed on the learning rate, which may explain its training instability and sensitivity to optimizer parameters. In § 4.3, Appendix B.2 and Appendix E, we also provide a similar analysis for the popular game optimization algorithms mentioned above. We emphasize however that while some of our results may be of independent interest for learning in general games, *our focus is DAL*. § 4.3 and § 6 show both theoretically and experimentally that the limitations mentioned above disappear if standard optimizers are replaced with ODE solvers of at least second order. These are straightforward to implement as drop-in replacements to existing optimizers. They also lead to more stable algorithms, allow for more aggressive learning rates and provide notable performance gains.

## 2 PRELIMINARIES

We focus on the UDA scenario and follow the formulation from Acuna et al. (2021). This makes our analysis general and applicable to most state-of-the-art DAL algorithms (e.g., Ganin et al. (2016); Saito et al. (2018); Zhang et al. (2019)). We assume that the learner has access to a source dataset (S) with *labeled* examples and a target dataset (T) with *unlabeled* examples, where the source inputs $x_i^s$ are sampled i.i.d. from a (source) distribution $P_s$ and the target inputs $x_i^t$ are sampled i.i.d. from a (target) distribution $P_t$, both over $\mathcal{X}$. We have $\mathcal{Y} = \{0, 1\}$ for binary classification, and $\mathcal{Y} = \{1, ..., k\}$ in the multiclass case. The risk of a hypothesis $h : \mathcal{X} \to \mathcal{Y}$ w.r.t. the labeling function $f$, using a loss function $\ell : \mathcal{Y} \times \mathcal{Y} \to \mathbb{R}_+$ under distribution $\mathcal{D}$ is defined as: $R_{\mathcal{D}}^\ell(h, f) := \mathbb{E}_{\mathcal{D}}[\ell(h(x), f(x))]$. For simplicity, we define $R_S^\ell(h) := R_{P_s}^\ell(h, f_s)$ and $R_T^\ell(h) := R_{P_t}^\ell(h, f_t)$. The hypothesis class of $h$ is denoted by $\mathcal{H}$.

Figure 1: We study domain-adversarial training from a game perspective. In DAL (Ganin et al. (2016)), three networks interact with each other: the feature extractor ($g$), the domain classifier ($\hat{h}'$) and the classifier ($\hat{h}$). During backpropagation, the GRL *flips* the sign of the gradient with respect to $g$.

**UDA** aims to minimize the risk in the target domain while only having access to labeled data in the source domain. This risk is upper bounded in terms of the risk of the source domain, the discrepancy between the two distributions and the joint hypothesis error $\lambda^*$:

**Theorem 1.** *(Acuna et al. (2021)) Let us note $\ell : \mathcal{Y} \times \mathcal{Y} \to [0, 1]$, $\lambda^* := \min_{h \in \mathcal{H}} R_S^\ell(h) + R_T^\ell(h)$, and $D_{h,\mathcal{H}}^\phi(P_s || P_t) := \sup_{h' \in \mathcal{H}} |\mathbb{E}_{x \sim P_s}[\ell(h(x), h'(x))] - \mathbb{E}_{x \sim P_t}[\phi^*(\ell(h(x), h'(x)))]|$. We have:*

$$R_T^\ell(h) \leq R_S^\ell(h) + D_{h,\mathcal{H}}^\phi(P_s || P_t) + \lambda^*. \tag{1}$$

The function $\phi : \mathbb{R}_+ \to \mathbb{R}$ defines a particular $f$-divergence and $\phi^*$ is its (Fenchel) conjugate. As is typical in UDA, we assume that the hypothesis class is complex enough and both $f_s$ and $f_t$ are similar in such a way that the non-estimable term ($\lambda^*$) is negligible and can be ignored.

**Domain-Adversarial Training** (see Figure 1) aims to find a hypothesis $h \in \mathcal{H}$ that jointly minimizes the first two terms of Theorem 1. To this end, the hypothesis $h$ is interpreted as the composition of $h = \hat{h} \circ g$ with $g : \mathcal{X} \to \mathcal{Z}$, and $\hat{h} : \mathcal{Z} \to \mathcal{Y}$. Another function class $\hat{\mathcal{H}}$ is then defined to formulate $\mathcal{H} := \{\hat{h} \circ g : \hat{h} \in \hat{\mathcal{H}}, g \in \mathcal{G}\}$. The algorithm tries to find the function $g \in \mathcal{G}$ such that $\hat{h} \circ g$ minimizes the risk of the source domain (i.e. the first term in Theorem 1), and its composition with $\hat{h}$ and $\hat{h}'$ minimizes the divergence of the two distributions (i.e. the second term in Theorem 1).

Algorithmically, the computation of the divergence function in Theorem 1 is estimated by a so-called *domain classifier* $\hat{h}' \in \hat{\mathcal{H}}$ whose role is to detect whether the datapoint $g(x_i) \in \mathcal{Z}$ belongs to the source or to the target domain. When there does not exist a function $\hat{h}' \in \hat{\mathcal{H}}$ that can properly distinguish between $g(x_i^s)$ and $g(x_i^t)$, $g$ is said to be invariant to the domains.

Learning is performed using GD and the GRL (denoted by $R_\lambda$) on the following objective:

$$\min_{\hat{h} \in \hat{\mathcal{H}}, g \in \mathcal{G}, \hat{h}' \in \hat{\mathcal{H}}} \mathbb{E}_{x \sim p_s}[\ell(\hat{h} \circ g, y)] - \alpha d_{s,t}(\hat{h}, \hat{h}', R_\lambda(g)), \tag{2}$$

where $d_{s,t}(\hat{h}, \hat{h}', g) := \mathbb{E}_{x \sim p_s}[\hat{\ell}(\hat{h}' \circ g, \hat{h} \circ g)] - \mathbb{E}_{x \sim p_t}[(\phi^* \circ \hat{\ell})(\hat{h}' \circ g, \hat{h} \circ g)]$. Mathematically, the GRL $R_\lambda$ is treated as a "pseudo-function" defined by two (incompatible) equations describing its forward and back-propagation behavior (Ganin & Lempitsky, 2015; Ganin et al., 2016). Specifically,

$$R_\lambda(x) := x \quad \text{and} \quad dR_\lambda(x)/dx := -\lambda, \tag{3}$$

where $\lambda$ and $\alpha$ are hyper-parameters that control the tradeoff between achieving small source error and learning an invariant representation. The surrogate loss $\ell : \mathcal{Y} \times \mathcal{Y} \to \mathbb{R}$ (e.g., cross-entropy) is used to minimize the empirical risk in the source domain. The choice of function $\hat{\ell} : \mathcal{Y} \times \mathcal{Y} \to \mathbb{R}$ and of conjugate $\phi^*$ of the $f$-divergence defines the particular algorithm (Ganin et al., 2016; Saito et al., 2018; Zhang et al., 2019; Acuna et al., 2021). From eq. 2, we can notice that GRL introduces an adversarial scheme. We next interpret eq. 2 as a three-player game where the players are $\hat{h}, \hat{h}'$ and $g$, and study its continuous gradient dynamics.

# 3 A GAME PERSPECTIVE ON DAL

We now interpret DAL from a game-theoretical perspective. In § 3.1, we rewrite the DAL objective as a three-player game. In this view, each of the feature extractor and two classifiers is a player. This allows us to define optimality in terms of local Nash Equilibrium (see Def. 2 in Appendices). In § 3.2, we introduce the vector field, the game Hessian and the tools that allow us to characterize local NE for the players. This characterization leads to our analysis of the continuous dynamics in § 4.

## 3.1 DOMAIN-ADVERSARIAL GAME

We now rewrite and analyze the DAL problem in eq. 2 as a three-player game. Let $\hat{\mathcal{H}}$, $\hat{\mathcal{H}}'$ and $\mathcal{G}$ be classes of neural network functions and define $\omega_1 \subseteq \Omega_1$ and $\omega_3 \subseteq \Omega_3$ as a vector composed of the parameters of the classifier and domain classifier networks $\hat{h} \in \hat{\mathcal{H}}$ and $\hat{h}' \in \hat{\mathcal{H}}$, respectively. Similarly, let $\omega_2 \subseteq \Omega_2$ be the parameters of the feature extractor network $g \in \mathcal{G}$. Their joint domain is denoted by $\Omega = \Omega_1 \times \Omega_2 \times \Omega_3$ and the joint parameter set is $\omega = (\omega_1, \omega_2, \omega_3)$. Let each neural network be a *player* and its parameter choice to be its individual strategy (here continuous). The goal of each player is then to selfishly minimize its own cost function $J_i : \Omega \to \mathbb{R}$. We use the subscript $_{-i}$ to refer to all parameters/players but $i$. With the notation introduced, we can now formally define the **Domain-Adversarial Game** as the three-player game $\mathcal{G}(\mathcal{I}, \Omega_i, J_i)$ where $\mathcal{I} := \{1, 2, 3\}$, $\dim(\Omega) = \sum_{i=1}^3 \dim(\Omega_i) = d, \Omega_i \subseteq \mathbb{R}^{d_i}$ and:

$$\begin{aligned} J_1(\omega_1, \omega_{-1}) &:= \ell(\omega_1, \omega_2) + \alpha d_{s,t}(\omega) \\ J_2(\omega_2, \omega_{-2}) &:= \ell(\omega_1, \omega_2) + \alpha\lambda d_{s,t}(\omega) \\ J_3(\omega_3, \omega_{-3}) &:= -\alpha d_{s,t}(\omega), \end{aligned} \tag{4}$$

We use the shorthand $\ell(\omega_1, \omega_2)$ for $\mathbb{E}_{x,y \sim p_s}[\ell(\omega_1 \circ \omega_2(x), y)]$, and $\omega_i$'s refer to the feature extractor $g$ and the classifiers ($\hat{h}$ and $\hat{h}'$). Similar notation follows for $d_{s,t}$. Here, we assume that each $J_i$ is smooth in each of its arguments $\omega_i \in \Omega_i$.

The gradient field of Equation (2) and the game's vector field (see § 3.2) are equivalent, making the original interpretation of DAL and our three-player formulation equivalent. However, it is worth noting that our intepretation does not explicitly require the use of $R_\lambda$ in $d_{s,t}$ in Equation (4). We can write optimality conditions of the above problem through the concept of Nash Equilibrium:

**Definition 1.** *(Nash Equilibrium (NE))* A point $\omega^* \in \Omega$ is said to be a Nash Equilibrium of the Domain-Adversarial Game if $\forall i \in \{1, 2, 3\}, \forall \omega_i \in \Omega_i$, we have: $J_i(\omega_i^*, \omega_{-i}^*) \leq J_i(\omega_i, \omega_{-i}^*)$.

In our scenario, the losses are not convex/concave. NE then does not necessarily exist and, in general, finding NE is analogous to, but much harder than, finding global minima in neural networks – which

is unrealistic using gradient-based methods (Letcher et al., 2019). Thus, we focus on local NE which relaxes the NE to a local neighborhood $\mathcal{B}(w^*, \delta) := \{||w - w^*|| < \delta\}$ with $\delta > 0$ (see Definition 2).

Intuitively, a NE means that no player has the incentive to change its own strategy (here parameters of the neural network) because it will not generate any additional pay off (here it will not minimize its cost function). We emphasize that each player only has access to its own strategy set. In other words, the player $J_1$ cannot change the parameters $\omega_2, \omega_3$. It only has access to $\omega_1 \in \Omega_1$.

While the motivation of the three-player game follows naturally from the original formulation of DAL where three networks interact with each other (see Figure 1), the optimization problem (2) could also be interpreted as the minimax objective of a two-player zero-sum game. Thus, a natural question arises: *can we interpret the domain-adversarial game as a two player zero-sum game?* This can be done for example by defining $\omega_{12}^* := (\omega_1^*, \omega_2^*)$, and considering the cost of the two players $(\omega_{12}, \omega_3)$ as $J_{12} = J$ and $J_3 = -J$ where $J(\omega_{12}, \omega_3) := \mathbb{E}_{p_s}[\ell(\omega_1, \omega_2)] + d_{s,t}(\omega)$. In general, however, *the solution of the two-player game $(\omega_{12}^*, \omega_3^*)$ is not equal to the NE solution of the three-player game* $(\omega_1^*, \omega_2^*, \omega_3^*)$. This is because the team optimal solution $\omega_{12}^* \neq (\omega_1^*, \omega_2^*)$ in general. We illustrate this in the following counterexample (see Başar & Olsder (1998) for more details):

**Example 1.** *Let the function* $J(\omega) := \frac{1}{2}\left(\omega_1^2 + 4\omega_1\omega_2 + \omega_2^2 - \omega_3^2\right)$.

*(a) Suppose the three-player game* $\omega = (\omega_1, \omega_2, \omega_3)$ *with* $J_1 = J_2 = J$ *and* $J_3 = -J$. *Each* $J_i$ *is strictly convex in* $\omega_i$. *The NE solution of the game* $\omega^* = (0, 0, 0)$ *is unique.*

*(b) Suppose the two-player game* $\omega = (\omega_{12}, \omega_3)$ *with* $J_{12} = J$ *and* $J_3 = -J$. *The solution* $\omega^*$ *from (a) is **not a NE solution**. To see this, let* $\hat{\omega} := (-1, 1, 0)$. *We have that* $J_{12}(\hat{\omega}) = -1 < J_{12}(\omega^*) = 0$. *This contradicts Definition 1. One can verify that **there is no NE** in this two-player scenario.*

## 3.2 Characterization of the Domain-Adversarial Game

We now introduce the game's vector field (also called *pseudo-gradient*) and the pseudo-gradient's Jacobian. We also provide a characterization of local NE based on them (see § 3). These are the core concepts used in our analysis (§ 4). We first define the game's vector field $v(w)$, and its Jacobian $H(\omega)$ (also called the *game Hessian* (Letcher et al., 2019)):

$$v(\omega) := (\nabla_{\omega_1} J_1, \nabla_{\omega_2} J_2, \nabla_{\omega_3} J_3) \in \mathbb{R}^d, \quad H(\omega) := \nabla v(\omega) \in \mathbb{R}^{d \times d} \tag{5}$$

Note that the vector field $v(w)$ and the three-player formulation naturally capture the behavior introduced by the GRL in the original formulation. Specifically, $v(\omega)$ is identical to the gradient with respect to the parameters of the original DAL objective *with GRL* (Equation (2)). Therefore, in both cases the behavior of GD is identical. Assuming the same initial conditions, they will reach the same solution. This shows the equivalence between our perspective and the original DAL formulation. We emphasize that by equivalence, we mean the same dynamics, and the same intermediate and final solutions. Another fact worth emphasizing is that $H(\omega)$ is asymmetric. This is in contrast with the Hessian in supervised learning. Before proceeding with a characterization of local NE in terms of $v(w)$ and $H(\omega)$, we first define sufficient and necessary conditions for local NEs:

**Proposition 1.** *(Necessary condition). Suppose each $J_i$ is twice continuously differentiable in each $\omega_i$, any **local NE** $\omega^*$ satisfies: i) $\nabla_{\omega_i} J_i(\omega^*) = 0$ and ii) $\forall i \in \{1, 2, 3\}$, $\nabla^2_{\omega_i, \omega_i} J_i(\omega^*) \succeq 0$.*

**Proposition 2.** *(Sufficient condition). Suppose each $J_i$ is twice continuously differentiable in each $\omega_i$. $\omega_i^*$ is a **local NE** if i) $\nabla_{\omega_i} J_i(\omega^*) = 0$ and ii) $\forall i, \nabla^2_{\omega_i, \omega_i} J_i(\omega^*) \succ 0$.*

The necessary and sufficient conditions from Propositions 1 and 2 are reminiscent of conditions for local optimality in continuous optimization (Nocedal & Wright, 2006). Similar conditions were also proposed in Ratliff et al. (2016) where the sufficient condition defines the *differential Nash equilibrium*. We can now characterize a local NE in terms of $v(w)$ and $H(\omega)$:

**Proposition 3.** *(Strict Local NE) $w$ is a strict local NE if $v(w) = 0$ and $H(\omega) + H(\omega)^\top \succ 0$.*

The sufficient condition implies that the NE is structurally stable (Ratliff et al., 2016). Structural stability is important as it implies that slightly biased estimators of the gradient (e.g., due to sampling noise) will not have vastly different behaviors in neighborhoods of equilibria (Mazumdar et al., 2020). In the following, we focus on the strict local NE (i.e., $\omega^*$ for which Proposition 3 is satisfied).

## 4 Learning Algorithms

We defined optimality as the local NE and provided sufficient conditions in terms of the pseudo-gradient and its Jacobian. In this section, we assume the existence of the strict local NE (Prop. 3)

in the neighborhood of the current point (e.g., initialization), and analyze the continuous gradient dynamics of the Domain-Adversarial Game (eq. 4 and eq. 5). We show that given the sufficient conditions from Prop. 3, asymptotic convergence to a local NE is guaranteed through an application of Hurwitz condition (Khalil, 2002). Most importantly, we show that using GD with the GRL could violate those guarantees unless its learning rate is upper bounded (see Thm. 2 an Cor. 1). This is in sharp contrast with known results from supervised learning where the implicit regularization introduced by GD has been shown to be desirable (Barrett & Dherin, 2021). We also analyze the use of higher-order ODE solvers for DAL and show that the above restrictions are not required if GD is replaced with them. Finally, we compare our resulting optimizers with recently algorithms in the context of games.

Our algorithmic analysis is based on the continuous gradient-play dynamics and the derivation of the *modified* or *high-resolution ODE* of popular integrators (e.g., GD/Euler Method and Runge-Kutta). This type of analysis is also known in the numerical integration community as backward error analysis (Hairer et al., 2006) and has recently been used to understand the implicit regularization effect of GD in supervised learning (Barrett & Dherin, 2021). High resolution ODEs have also been used in Shi et al. (2018) to understand the acceleration effect of optimization algorithms, and more recently in Lu (2020). As in Shi et al. (2018); Lu (2020); Barrett & Dherin (2021), our derivation of the high resolution ODEs is in the full-batch setting. The derivation of the stochastic dynamics of stochastic discrete time algorithms is significantly more complicated and is beyond the scope of this work.

We experimentally demonstrate that our results are also valid when there is stochasticity due to sampling noise in the mini-batch. We emphasize that our analysis does not put any constraint or structure on the players' cost functions as opposed to Azizian et al. (2020); Zhang & Yu (2020). In our problem, the game is neither bilinear nor necessarily strongly monotone. See proofs in appendices.

## 4.1 CONTINUOUS GRADIENT DYNAMICS

Given $v(\omega)$ the continuous gradient dynamics can be written as:

$$\dot{\omega}(t) = -v(\omega). \tag{6}$$

For later reasons and to distinguish between eq. 6 and the gradient flow, we will refer to these as the gradient-play dynamics as in Başar & Olsder (1998); Mazumdar et al. (2020). These dynamics are well studied and understood when the game is either a potential or a purely adversarial game (see definitions in appendices). While eq. 2 may look like a single objective, the introduction of the GRL ($R_\lambda$), makes a fundamental difference between our case and the dynamics that are analyzed in the single-objective gradient-based learning and optimization literature. We summarize this below:

**Proposition 4.** *The domain-adversarial game is neither a potential nor necessarily a purely adversarial game. Moreover, its gradient dynamics are not equivalent to the gradient flow.*

Fortunately, we can directly apply the Hurwitz condition (Khalil, 2002) (also known as the condition for asymptotic stability, see Appendix A.1) to derive sufficient conditions for which the continuous dynamics of the gradient play would converge.

**Lemma 1.** *(Hurwitz condition) Let $\nabla v(w^*)$ be the Jacobian of the vector field at a stationary point $w^*$ where $v(w^*) = 0$. If the real part of every eigenvalue $\lambda$ of $\nabla v(w^*)$ (i.e. in the spectrum $Sp(\nabla v(w^*))$) is positive then the continuous gradient dynamics are asymptotically stable.*

In this work, we assume the algorithms are initialized in a neighborhood of a strict local NE $\omega^*$. Therefore, Lemma 1 provides sufficient conditions for the asymptotic convergence of the gradient-play dynamics to a local NE. In practice this assumption may not hold, and it is computationally hard to verify. Despite this, our experiments show noticeable performance gains in several tasks, benchmarks and network architectures (see § 6).

## 4.2 ANALYSIS OF GD WITH THE GRL

We showed above that given the existence of a strict local NE, the gradient-play dynamics are attracted to the strict local NE. A natural question then arises: *If under this assumption local asymptotic convergence is guaranteed, what could make DAL notoriously hard to train and unstable?* In practice, we do not have access to an explicit solution of the ODE. Thus, we rely on integration algorithms to approximate the solution. One simple approach is to use the Euler method:

$$w^+ = w - \eta v(w). \tag{7}$$

This is commonly known as GD. The equivalence between $v(w)$ (game's vector field) and the gradient of Equation (2) (original DAL formulation) follows from the use of the GRL ($R_\lambda$). We remind the reader that the GRL is a "pseudo-function" defined by two (incompatible) equations describing its forward and back-propagation behavior, i.e., a flip in the gradient's sign for the backward pass (see Figure 1, Section 2 and Ganin et al. (2016)). Equation (7) is then the default algorithm used in DAL. Now, to provide an answer to the motivating question of this section, we propose to analyze the high-resolution ODE of this numerical integrator (i.e., Euler) and in turn its asymptotic behavior. This is similar to deriving the modified continuous dynamics for which the integrator produces the exact solution (Hairer et al., 2006) and applying Hurwitz condition on the high-resolution ODE.

**Theorem 2.** *The high resolution ODE of GD with the GRL up to $O(\eta)$ is:*

$$\dot{w} = -v(w) - \frac{\eta}{2}\nabla v(w)v(w) \tag{8}$$

*Moreover, this is asymptotically stable (see Appendix A.1) at a stationary point $w^*$ (Proposition 3) iff for all eigenvalue written as $\lambda = a + ib \in Sp(-\nabla v(w^*))$, we have $0 > \eta(a^2 - b^2)/2 > a$.*

A striking difference between Equation (6) and Equation (8) is made clear (additional term marked in red). This additional term is a result of the discretization of the gradient-play dynamics using Euler's method (i.e. GD) and leads to a different Jacobian of the dynamics. This term was recently shown to be beneficial for standard supervised learning (Barrett & Dherin, 2021), where $\nabla v(\omega^*)$ is symmetric and thus only has real eigenvalues. In our scenario, this term is undesirable. In fact, this additional term puts an upper bound on the learning rate $\eta$. The following corollary formalizes this:

**Corollary 1.** *The high resolution ODE of GD with GRL in Equation (8) is asymptotically stable only if the learning rate $\eta$ is in the interval: $0 < \eta < \frac{-2a}{b^2 - a^2}$, for all $\lambda = a + ib \in Sp(-\nabla v(w^*))$ with large imaginary part (i.e., such that $|a| < |b|$).*

To have good convergence properties, the imaginary part of the eigenvalues of $-\nabla v(w^*)$ must be small enough. Therefore, if some eigenvalue $\lambda = a + ib$ satisfies $a < 0$ and $b^2 \gg a^2 \gg -2a$, the learning rates should be chosen to be very small. This is verified in Section 6 and in Example 2.

**Example 2.** *Consider the three-player game where $\ell(w_1, w_2) = w_1^2 + 2w_1w_2 + w_2^2$, $\lambda = 1$ and $d_{s,t}(w_2, w_3) = w_2^2 + 99w_2w_3 - w_3^2$. Then $\dot{w} = -v(w)$ becomes: $\dot{w} = Aw = \begin{pmatrix} -2 & -2 & 0 \\ -2 & -4 & -99 \\ 0 & 99 & -2 \end{pmatrix}$. The eigenvalues of $A$ are $-2$ and $-3 \pm 2i\sqrt{2449}$. From Corollary 1, $\eta$ should be $0 < \eta < 6.2 \times 10^{-3}$.*

### 4.3 HIGHER ORDER ODE SOLVERS

The limitation described above exists because GD with the GRL can be understood as a discretization of the gradient-play dynamics using Euler's Method. Thus, it only approximates the continuous dynamics up to $O(\eta)$. To obtain a better approximation, we consider Runge-Kutta (RK) methods of order two and beyond (Butcher, 1996). For example, take the improved Euler's method (a particular RK method of second order) that can be written as:

$$w^+ = w - \frac{\eta}{2}(v(w) + v(w - \eta v(w))). \tag{9}$$

Comparing Equation (9) (i.e., update rule of RK2) with Equation (7) (i.e., update rule of GD), one can see that the RK2 method is straightforward to implement in standard deep learning frameworks. Moreover, it does not introduce additional hyper-parameters. More importantly, such discrete dynamics approximate the continuous ODE of Equation (6) to a higher precision. In Appendix C, we provide asymptotic guarantees for the high resolution ODE of general RK methods, their generalized expression and the algorithm pseudo-code. See also PyTorch pseudo-code in Appendix E.

**Limitation.** A disadvantage of using high-order solvers is that they require additional extra steps. Specifically, one extra step in the case of RK2 (computation of the additional second term in Equation (9)). In our implementation, however, this was less than 2x slower in wall-clock time (see Appendix E.5 for more details and wall-clock comparison). Moreover, if not initialized in the neighborhood of a local NE, high-order solvers and gradient-based methods might also converge to a non-NE as described in Mazumdar et al. (2019) although this is likely a rare case.

**Comparison vs other game optimization algorithms.** DAL has not been previously interpreted from a game perspective. Our interpretation allows us to bring recently proposed algorithms to the context of differentiable games (Zhang & Yu, 2020; Azizian et al., 2020) to DAL. Classic examples are the Extra-Gradient (EG) method (Korpelevich, 1976) and Consensus Optimization (CO) (Mescheder et al., 2017). In Appendix B.2 we analyze the continuous dynamics of the EG method, and show that

we cannot take the learning rate of EG to be large either. Thus, we obtain a similar conclusion as Corollary 1. Then, in practice for DAL, stability for EG comes at the price of slow convergence due to the use of small learning rates. We experimentally show this in Figure 3. With respect to CO, we show in Appendix C that this algorithm can be interpreted in the limit as an approximation of the RK2 solver. In practice, if its additional hyper-parameter ($\gamma$) is tuned thoroughly, CO may approximate the continuous dynamics better than GD and EG. We believe this may be the reason why CO slightly outperforms GD and EG (see Appendix E.4). In all cases, RK solvers outperform GD, EG and CO. This is in line with our theoretical analysis since they better approximate the continuous dynamics (Hairer et al., 2006). It is worth noting that many other optimizers have recently been proposed in the context of games e.g., Gidel et al. (2019a); Hsieh et al. (2020); Lorraine et al. (2021a;b). Some of them are modifications of the EG method that we compared to e.g. Extra-Adam (Gidel et al., 2019a) or double step-size EG (Hsieh et al., 2020). More practical modifications in terms of adaptive step size could also be applied on top of RK solvers as done in Qin et al. (2020). A comparison of all existing game optimizers in DAL, and a better theoretical understanding of such modification on RK solvers are beyond the scope of this work. However, we believe it is an interesting and unexplored research direction that our game perspective on DAL enables.

## 5 RELATED WORK

To the best of our knowledge, DAL has not been previously analyzed from a game perspective. Moreover, the stability of the optimizer and the implications of introducing the GRL has not been analyzed either. Here, we compare our results with the general literature.

**Gradient-Based Learning in Games.** Ratliff et al. (2016) proposed a characterization of local Nash Equilibrium providing sufficient and necessary conditions for its existence. Mazumdar et al. (2020) proposed a general framework to analyze the limiting behavior of the gradient-play algorithms in games using tools from dynamical systems. Our work builds on top of this characterization but specializes them to the domain-adversarial problem. We propose a more stable learning algorithm that better approximates the gradient-play dynamics. Our resulting algorithm does not introduce explicit adjustments or modify the learning dynamics, nor does it require the computation of the several Hessian vector products or new hyperparameters. This is in contrast with general algorithms previously analyzed in the context of differentiable games (Azizian et al., 2020; Letcher et al., 2019).

**Integration Methods and ML.** Scieur et al. (2017) showed that accelerated optimization methods can be interpreted as integration schemes of the gradient flow equation. Zhang et al. (2018) showed that the use of high order RK integrators can achieve acceleration in convex functions. In the context of two-players game (i.e GANs), Gemp & Mahadevan (2018) consider using a second-order ODE integrator. More recently, Qin et al. (2020) proposed to combine RK solvers with regularization on the generators' gradient norm. Chen et al. (2018) interpreted the residual connection in modern networks as the Euler's integration of a continuous systems. In our case, we notice that the combination of GD with the GRL can be interpreted as the Euler's discretization of the continuous gradient play dynamics, which could prevent asymptotic convergence guarantees. We then study the discretization step of popular ODE solvers and provide simple guarantees for stability. Moreover, our analysis is based on a novel three-player game interpretation of the domain-adaptation problem. This is also different from a single potential function or two-player games (i.e. GANs).

**Two-Player Zero-Sum Games** have recently received significant attention in the machine learning literature due to the popularity of Generative Adversarial Networks (GANs) (Goodfellow et al., 2014). For example, several algorithms have been proposed and analyzed (Mescheder et al., 2017; Mertikopoulos et al., 2019; Gidel et al., 2019a;b; Zhang & Yu, 2020; Hsieh et al., 2020), in both deterministic and stochastic scenarios. In our problem, we have a general three-player games resulting of a novel game interpretation of the domain-adversarial problem. It is worth noting that while Gidel et al. (2019a) focused on GANs, their convergence theory and methods for stochastic variational inequalities could also be applied to three-players games and thus DAL using our perspective.

## 6 EXPERIMENTAL RESULTS

We conduct an extensive experimental analysis. We compare with default optimizers used in domain-adversarial training such as GD, GD with Nesterov Momentum (GD-NM) (as in Sutskever et al. (2013)) and Adam (Kingma & Ba, 2014). We also compare against recently proposed optimizers in the context of differentiable games such as EG (Korpelevich, 1976) and CO (Mescheder et al., 2017). We focus our experimental analysis on the original domain-adversarial framework of Ganin et al.

(2016) (DANN). However, in section 6.2, we also show the versatility and efficacy of our approach improving the performance of recently proposed SoTA DAL framework (e.g., $f$-DAL (Acuna et al., 2021) combined with Implicit Alignment (Jiang et al., 2020)).

## 6.1 EXPERIMENTAL ANALYSIS ON DIGITS

**Implementation Details.** Our first experimental analysis is based on the digits benchmark with models trained from scratch (i.e., with random initialization). This benchmark constitutes of two digits datasets **MNIST** (CC BY-SA 3.0) and **USPS** (LeCun et al., 1998; Long et al., 2018) with two transfer tasks (M → U and U → M). We adopt the splits and evaluation protocol from Long et al. (2018) and follow their standard implementation details.

For GD-NM, we use the default momentum value (0.9). We follow the same approach for the additional hyper-parameters of Adam. Hyperparameters such as learning rate, learning schedule and adaptation coefficient ($\lambda$) are determined for all optimizers by running a dense grid search and selecting the best hyper-parameters on the transfer task M→U. As usual in UDA, the best criteria are determined based on best transfer accuracy. The same parameters are then used

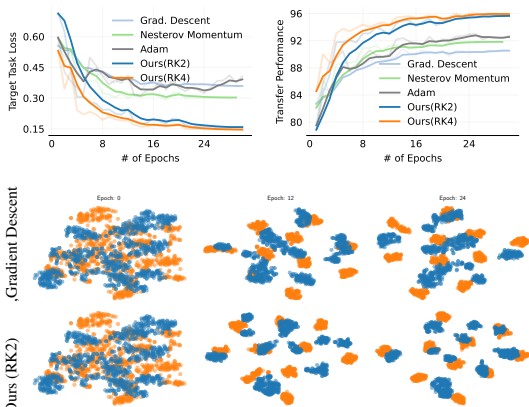

Figure 2: Our method vs popular optimizers on the Digits Benchmark. (**Top-Left**) Loss in target domain. (**Top-Right**) Transfer performance. (**Bottom**) $t$-SNE Visualization of the last layer representations during training. Our method converges faster, has better performance and produces more aligned features faster.

for the other task (i.e., U→M). We use the same seed and identically initialize the network weights for all optimizers. This analysis is conducted on Jax (Bradbury et al., 2018) (see Appendix D).

**Comparison vs optimizers used in DAL.** Figure 2 (top) illustrates the training dynamics for the loss in the target domain and the performance transfer. As expected, our optimizer converges faster and achieves noticeable performance gains. A core idea of DAL is to learn domain-invariant representations, thus we plot in Figure 2 (bottom) $t$-SNE (Van der Maaten & Hinton, 2008) visualizations of the last layer features of the network. We show this over a sequence of epochs for GD with GRL vs RK2. A different color is used for the source and target datasets. In the comparison vs Adam, we emphasize that Adam computes adaptive learning rates which our method does not. That said, Figure 2 shows that our two methods RK2 and RK4

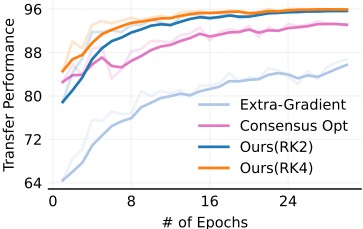

Figure 3: Comparison among optimization algorithms (M→U) with DANN.

outperform all baselines in terms of both convergence and transfer performance. In Figure 7, we show how unstable these standard optimizers are when more aggressive step sizes are used. This is in line with our theoretical analysis. Experimentally, it can be seen that in DAL, GD is more stable than GD-NM and Adam, with the latter being the most unstable. This sheds lights on why well tuned GD-NM is often preferred over Adam in DAL.

**Comparison vs game optimization algorithms.** We now compare RK solvers vs other recently proposed game optimization algorithms. Specifically, we compare vs the EG method (Korpelevich, 1976) and CO (Mescheder et al., 2017). In every case, we perform a dense grid under the same budget for all the optimizer and report the best selection (see Appendix E for details). In line with our theoretical analysis of the continuous dynamics of the EG, we notice that the EG method is not able to train with learning rates bigger than 0.006, as a result it performs signficantly worse than any other optimizer (including simple GD). Also inline with our theoretical analysis, CO performs better than EG and all other popular gradient algorithms used in DAL. This is because CO can be seen as an approximation of Heun's Method (RK2). More details on this in supplementary.

**Robustness to hyper-parameters.** Figure 4 shows transfer performance of our method for different choices of hyper-parameters while highlighting (green line) the best score of the best performing GD hyperparameters on the same dataset. Our method is robust to a wide variety of hyperparameters.

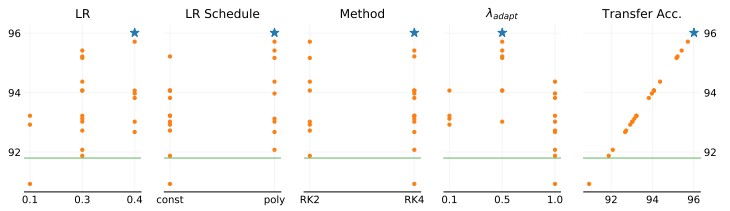 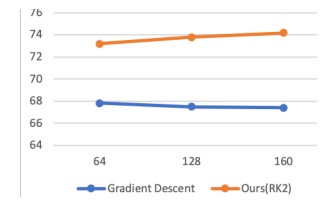

Figure 4: *Robustness to hyperparameters*. We compare the transfer performance of our method for different hyperarameters in the task M→ U in the Digits benchmark. Green line shows the best score for the best performing hyperparameters of GD. Blue star corresponds to the best solution. Our method performs well for a wide variety of hyperparameters.

Figure 5: *Sensitivity to Sampling Noise* . Different amounts of sampling noise controlled by the batch size (64, 128, 160) (Visda).

| Method | M→U | U→M | Avg |
|---|---|---|---|
| GD | $90.0 \pm 0.4$ | $93.4 \pm 0.7$ | 91.7 |
| Adam | $92.8 \pm 0.3$ | $96.8 \pm 0.2$ | 94.8 |
| GD-NM | $91.8 \pm 0.3$ | $94.4 \pm 0.4$ | 93.1 |
| Ours(RK2) | $\mathbf{95.1} \pm 0.1$ | $\mathbf{97.5} \pm 0.2$ | **96.3** |
| Ours(RK4) | $\mathbf{95.0} \pm 1.4$ | $97.3 \pm 0.1$ | 96.1 |

Table 3: Accuracy (%) on Digits (DANN).

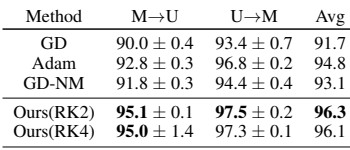

Figure 6: Transfer Performance on Visda (DANN).

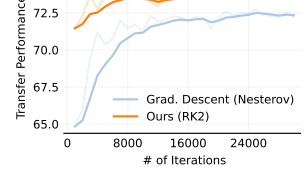

Figure 7: *Stability anal. on Digits*. Most aggressive step size before divergence. Adam diverges for $\eta > 0.001$.

### 6.2 COMPARISON IN COMPLEX ADAPTATION TASKS

We evaluate the performance of our algorithm with Resnet-50 (He et al., 2016) on more challenging adaptation benchmarks. Specifically, this analysis is conducted on Visda-2017 benchmark (Peng et al., 2017). This is a simulation-to-real dataset with two different domains: (S) synthetic renderings of 3D models and (R) real images. For this experiment, we use PyTorch (Paszke et al., 2019), our evaluation protocol follows Zhang et al. (2019) and uses ResNet-50 as the backbone network. For the optimizer parameters, we tune

| Method | Sim→Real |
|---|---|
| GD-NM | $71.7 \pm 0.7$ |
| Ours(RK2) | $\mathbf{73.8} \pm 0.3$ |

Table 1: Accuracy (DANN) on Visda 2017 with ResNet-50.

thoroughly GD-NM, which is the optimizer used in this setting (Long et al., 2018; Zhang et al., 2019; Jiang et al., 2020; Acuna et al., 2021). For ours, we keep the hyper-parameters, but increase the learning rate (0.2), and the batch size to 128. In this task, our approach corresponds to the improved Euler's method (RK2). Table 1 shows the comparison. Figure 6 compares the training dynamics of our method vs GD-NM. In Figure 5, we evaluate the sensitivity of our method (in terms of transfer performance) to sampling noise as controlled by the batch size.

**Improving SoTA DAL frameworks.** We use this complex visual adaptation task to showcase the applicability of our method to SoTA DAL frameworks. Specifically, we let the DA method being $f$-DAL Pearson as in Acuna et al. (2021) with Implicit Alignment Jiang et al. (2020). We use the tuned parameters and optimizer from Acuna et al. (2021); Jiang et al.

| Method | Sim→Real |
|---|---|
| $f$-DAL - GD-NM | 72.9 (29.5K iter) |
| $f$-DAL - RK2 **(Ours)** | **76.4** (10.5K iter) |

Table 2: Comparison using SoTA DA adversarial frameworks with ResNet-50 on Visda.

(2020) as the baseline. In our case, we only increase the learning rate (0.2). Table 2 shows that our method achieves peak results (**+3.5%**) in **10.5K** iterations (vs **29.5K** iterations for GD-NM).

**Natural Language Processing Tasks.** We also evaluate our approach on natural language processing tasks on the Amazon product reviews dataset (Blitzer et al., 2006). We show noticeable gains by replacing the GD with either RK2 or RK4. Results and details can be found in Appendix E.1.

## 7 CONCLUSIONS

We analyzed DAL from a game-theoretical perspective where optimality is defined as local NE. From this view, we showed that standard optimizers in DAL can violate the asymptotic guarantees of the gradient-play dynamics, requiring careful tuning and small learning rates. Based on our analysis, we proposed to replace existing optimizers with higher-order ODE solvers. We showed both theoretically and experimentally that these are more stable and allow for higher learning rates, leading to noticeable improvements in terms of the transfer performance and the number of training iterations. We showed that these ODE solvers can be used as a drop-in replacement and outperformed strong baselines.

**Acknowledgements.** We would like to thank James Lucas, Jonathan Lorraine, Tianshi Cao, Rafid Mahmood, Mark Brophy and the anonymous reviewers for feedback on earlier versions of this work.

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

## SUPPLEMENTARY MATERIAL

## CONTENTS

# A    CONCEPTS IN GAME THEORY

## A.1    DEFINITIONS

**Definition 2.** *(Local Nash Equilibrium) : A point $(w_i^*, w_{-i}^*) \in \Omega$ is said to be a local Nash Equilibrium of the domain-adversarial game if there exists some $\delta > 0$ such that:*

$$\forall i \in \{1, 2, 3\}, \quad J_i(w_i^*, w_{-i}^*) \leq J_i(w_i, w_{-i}^*), \quad \text{s.t. } ||\omega_i - \omega_i^*||_2 < \delta \tag{10}$$

Intuitively, this is restricting the concept of NE to a local neighborhood $\mathcal{B}(x^*, \delta) := \{||x - x^*||_2 < \delta\}$ with $\delta > 0$.

A more practical characterization of the NE can be given in terms of the Best Response Map of each player which we now define.

**Definition 3.** *(Best Response Map (BR)) The best response map $BR_i : \Omega_{-i} \rightrightarrows \Omega_i$ of player $i$ is defined as:*

$$BR_i(\omega_{-i}) := \underset{\omega_i \in \Omega_i}{\arg\min} \, J_i(\omega_i, \omega_{-i}), \tag{11}$$

here the symbol $\rightrightarrows$ emphasizes that the best response map is generally a set map and not a singleton, thus it is not a function in general. In other words, there may be a subset of element in $\Omega_i$ for which $J_i(., \omega_{-i})$ is a minimum.

The notion of NE can be defined in terms of the *generalized* BR $: \Omega \rightrightarrows \Omega$ map. This can be thought as an stacked vector where the $i$-th element of BR is $BR_i(\omega_{-i})$.

**Proposition 5.** *A point $w_i^* \in \Omega$ is said to be a NE of the game if it is a fixed point of the generalized BR $: \Omega \rightrightarrows \Omega$ map. That is,*

$$\omega^* \in BR(\omega^*) \implies \forall i \in \{1, 2, 3\}, \quad \omega_i^* \in BR_i(\omega_{-i}^*) \tag{12}$$

*Proof.*  This follows from the definitions of BR map and NE.                                              $\square$

**Definition 4.** *(Asymptotically Stable) A point $\omega$ is said to be a locally asymptotically stable point of the continuous dynamics $\dot{\omega} = f(\omega)$ if $Re(\lambda) < 0$ for all $\lambda \in Sp(\nabla f(\omega))$, where $Sp(\nabla f(\omega))$ is the spectrum of $\nabla f(\omega)$.*

Definition 4 is also known as the Hurwitz condition Khalil (2002).

**Definition 5.** *A stationary point $x$ of a $C^2$ function $\phi : \mathbb{R}^n \to \mathbb{R}$ is said to be a strict saddle point if:*

- $\lambda_{\min}(\nabla_{xx}^2 \phi(x^*)) < 0$ *and,*

- $\lambda(\nabla_{xx}^2 \phi(x^*)) > 0$, *for any other $\lambda \in sp(\nabla_{xx}^2 \phi(x^*))$*

## A.2    GAMES CHARACTERIZATIONS

**Potential Games.** Potential Games were introduced in Monderer & Shapley (1996) and can be defined as a type of game for which there exists an implicit *potential* function $\phi : \Omega \to R$ such that $\nabla\phi(\omega) = v(\omega)$. Consequently, a necessary and sufficient condition for the game to be potential is the Jacobian of the vector field $\nabla v(\omega)$ being symmetric (see 3.3 in Mazumdar et al. (2020) and Monderer & Shapley (1996)).

**Purely Adversarial Games.** This particular type of game refers to the other extreme in which $H(\omega)$ is a non-symmetric matrix with purely imaginary eigenvalues. If the game Hessian is skew-symmetric these have also been called Hamiltonian Games Letcher et al. (2019).

## A.3    CASE OF STUDY IN DANN. ORIGINAL FORMULATION FROM GANIN ET AL. (2016)

As mentioned in the main text (Section 2), our analysis is compatible with both the original and more recent formulation of domain-adversarial training such as Zhang et al. (2019); Acuna et al. (2021). In this section, we specifically derive additional results for DANN Ganin et al. (2016).

In order to obtain the original formulation of DANN, let us define $\hat{\ell}(\_, b) = \log(\sigma(b))$ and $\phi^*(t) = -\log(1 - e^t)$ in Equation (2). This corresponds to the Jensen-Shannon divergence (JS) (up to a constant shift that does not affect optimization). We can then rewrite $d_{s,t}$ as:

$$d_{s,t} = \mathbb{E}_{x \sim p_s} \log \sigma \circ \hat{h}' \circ g(x) + \mathbb{E}_{x \sim p_t} \log(1 - \sigma \circ \hat{h}' \circ g(x)) \tag{13}$$

where $\sigma(x) := \frac{1}{1 + e^{-x}}$. To simplify the notation, we write $\mathcal{H} := \sigma \circ \mathcal{H}$.

We can now re-define the pseudo-gradient $v(w)$ of the game as the gradient of each player loss with respect to its parameters. Letting $\alpha = 1$, we get from Equation (4).

$$v(\omega) := (\nabla_{\omega_1} \ell, \nabla_{\omega_2}(\ell + \lambda d_{s,t}), -\nabla_{\omega_3} d_{s,t}) \in \mathbb{R}^d. \tag{14}$$

The following propositions characterize local NE in terms of the pseudo-gradient $v(w)$ and its Jacobian $H(\omega)$.

**Proposition 6.** *(Local NE)* Suppose $v(w) = 0$ and

$$\begin{pmatrix} \nabla^2_{\omega_1} \ell & \nabla^2_{\omega_1, \omega_2} \ell \\ \nabla^2_{\omega_1, \omega_2} \ell & \nabla^2_{\omega_2}(\ell + \lambda d_{s,t}) \end{pmatrix} \succ 0, \ \nabla^2_{\omega_3} d_{s,t} \prec 0, \tag{15}$$

*then $w$ is an isolated local NE.*

The proof is simple and follows from Propositions 1 and 2, the definition of the vector field $v(\omega)$ and the condition $H + H^\top \succ 0$.

**Cooperation with Competition**. By examining the matrix $H(\omega)$, one can see that, in our scenario, the game is neither a potential game nor a purely adversarial game. However, we can write the vector field in the following form:

$$v(w) = \underbrace{\begin{pmatrix} \nabla_{\omega_1} \ell \\ \nabla_{\omega_2} \ell \\ 0 \end{pmatrix}}_{\nabla \phi(\omega)} + \underbrace{\begin{pmatrix} 0 \\ \lambda \nabla_{\omega_2} d_{st} \\ -\nabla_{\omega_3} d_{st} \end{pmatrix}}_{\hat{v}(\omega)} \tag{16}$$

where the first part corresponds to the gradient of the potential function $\phi(\omega) = \ell(\omega_1, \omega_2)$. The second part, on the other hand, corresponds to a function $\hat{v}(w)$ whose Jacobian is a non-symmetric matrix. Analyzing them separately leads to either a potential or an adversarial game respectively. We define this particular type of game as **cooperation** (i.e., in the potential term) with **competition** (i.e., the adversarial term).

It is worth noting that, while the spectrum of the game Hessian for the first term has only real eigenvalues, the second term can have complex eigenvalues with a large imaginary component. Indeed, it can be shown that this second term approximates the one obtained for a GAN using the non-saturating loss proposed by Goodfellow et al. (2014) (e.g. $\lambda = 1$). In other words, the second term can be written as the pseudo-gradient of the two player zero-sum game $\min_{\omega_2} \max_{\omega_3} d_{st}$. Building on this key observation and the work of Mescheder et al. (2017); Berard et al. (2020) (Figure 4), where it was experimentally shown that the eigenvalues of the game Hessian for GANs have indeed a large imaginary component around stationary points, we can assume that the spectrum of the game Hessian in our case also have eigenvalues with a large imaginary component around the stationary points. This observation can also be used with Corollary 1 to further motivate the use of higher-order ODE solvers instead of GD with the GRL.

**Example 3.** *Consider the three-player game Equation (16) where $\ell(w_1, w_2) = w_1^2 + 2w_1 w_2 + w_2^2$, $\lambda = 1$ and $d_{s,t}(w_2, w_3) = w_2^2 + 99w_2 w_3 - w_3^2$. The gradient play dynamics $\dot{w} = -v(w)$ becomes:*

$$\dot{w} = Aw = \begin{pmatrix} -2 & -2 & 0 \\ -2 & -4 & -99 \\ 0 & 99 & -2 \end{pmatrix} w.$$

*The eigenvalues of $A$ are $-2$ and $-3 \pm 2i\sqrt{2449}$. From Corollary 1, $\eta$ should be $0 < \eta < 6.2 \cdot 10^{-3}$.*

**Is the three-player game formulation desired?** In domain adaptation, optimization is *a means to an end*. The final goal is to minimize the upper bound from Theorem 1 to ensure better performance in the target domain. One might then wonder whether interpreting optimality in terms of NE is desirable. In our context, NE means finding the optimal $g^*, \hat{h}^*$ and $\hat{h}'^*$ of the cost functions defined in Equation (4). This in turns leads to minimizing the upper bound in Theorem 1.

**Remark on sequential games**: Recently, Jin et al. (2020) introduced a notion of local min-max optimality for two-player's game exploiting the sequential nature of some problems in adversarial ML (i.e GANs). In domain-adversarial learning, updates are usually performed in a simultaneous manner using the GRL. Thus, we focus here on the general case where the order of the players is not known.

# B  DERIVATION OF HIGH-RESOLUTION ODES

**Lemma 2.** *The high resolution ODE of resulting of the GD algorithm with the GRL is:*

$$\dot{w} = -v(w) - \frac{\eta}{2}\nabla v(w)v(w) + O(\eta^2), \tag{17}$$

*Proof.* This follows from Corollary 1 of Lu (2020). □

## B.1  HIGH-RESOLUTION ODE OF SECOND-ORDER RUNGE–KUTTA METHOD

The high-resolution ODE was discussed in Shi et al. (2018); Lu (2020). For discrete algorithms with the following update:

$$w^+ = w + f(\eta, w), \tag{18}$$

we can think of the trajectory as a discretization of the continuous dynamics $w : [0, +\infty) \to \mathbb{R}^d$, and in Equation (18), we have $w = w(t)$, $w^+ = w(t + \eta)$. Here, with slight abuse of notation we also use $w$ for the continuous function of dynamics.

We derive high-resolution ODE of the second-order Runge–Kutta method:

$$w_{k+1/2} = w_k - \frac{\eta}{2\alpha}v(w_k), \ w_{k+1} = w_k - \eta((1-\alpha)v(w_k) + \alpha v(w_{k+1/2})),$$

where $0 < \alpha \leq 1$ and $\alpha$ is a constant. If $\alpha = 1/2$, we obtain Heun's method; if $\alpha = 1$, we obtain the midpoint method; if $\alpha = 2/3$, we obtain the Ralston's method. Combining the two equations, we have:

$$\frac{w_{k+1} - w_k}{\eta} = -(1-\alpha)v(w_k) - \alpha v(w_k - \frac{\eta}{2\alpha}v(w_k)). \tag{19}$$

Using the Taylor expansion:

$$v(w_k - \frac{\eta}{2\alpha}v(w_k)) = v(w_k) - \frac{\eta}{2\alpha}\nabla v(w_k)^\top v(w_k) + O(\eta^2)$$

Plugging it back into Equation (19) and using the Taylor expansion $w_{k+1} = w_k + \eta\dot{w} + \eta^2\ddot{w}/2$, we have:

$$\dot{w} + \frac{1}{2}\eta\ddot{w} = -v(w) + \frac{1}{2}\nabla v(w)^\top v(w) + O(\eta^2). \tag{20}$$

Now we make the assumption that we have the high-resolution ODE that:

$$\dot{w} = f_0(w) + \eta f_1(w) + O(\eta^2). \tag{21}$$

Taking the derivative over $t$ we have:

$$\ddot{w} = \nabla f_0(w)f_0(w) + O(\eta). \tag{22}$$

Combining Equation (20), Equation (21) and Equation (22), we obtain that:

$$f_0(w) = -v(w), f_1(w) = 0, \tag{23}$$

i.e., the high resolution ODE of second-order Runge–Kutta method is:

$$\dot{w} = -v(w) + O(\eta^2). \tag{24}$$

## B.2  CONTINUOUS DYNAMICS OF EXTRA-GRADIENT (EG)

The continuous dynamics of Gradient Descent Ascent (GDA), Extra-Gradient (EG) and Heun's method can be summarized as follows:

$$\dot{w} = v(w) + \alpha\nabla v(w)v(w)$$

For GDA, we have $\alpha = -\eta/2$; for EG, we have $\alpha = \eta/2$ (Lu, 2020); for Heun's method, $\dot{w} = v(w) + O(\eta^2)$. The Jacobian of the dynamics at the stationary point is $\nabla v(w) + \alpha\nabla v(w)^2$. Take $\lambda = a + ib \in Sp(\nabla v(w))$. The eigenvalue of the Jacobian of the dynamics is:

$$\alpha(a + ib)^2 + a + ib = a + \alpha(a^2 - b^2) + i(b + 2ab)\alpha. \tag{25}$$

We want the real part to be negative, i.e.:

$$a + \alpha(a^2 - b^2) < 0, \tag{26}$$

and thus:

$$a(1 + \alpha a) < \alpha b^2. \tag{27}$$

for EG, $\alpha = \eta/2$ and the dynamics diverges if $a(1+(\eta/2)a) \geq \eta b^2/2$. When $\eta$ is large, and $\eta(a^2-b^2)/2 \geq -a$ then it diverges. However, the high-resolution ODE of second-order Runge–Kutta methods only requires $a < 0$.

### B.3 High-resolution ODE of classic fourth-order Runge–Kutta method (RK4)

In this subsection, we derive the high-resolution ODE of the classic fourth-order Runge–Kutta method. We prove the following result:

**Theorem 3.** *The high-resolution ODE of the classic fourth-order Runge–Kutta method (RK4):*

$$w^+ = w - \frac{\eta}{6}(v(w) + 2v_2(w) + 2v_3(w) + v_4(w)), \tag{28}$$

*where*

$$v_2(w) = v(w - \frac{\eta}{2}v(w)), \; v_3(w) = v(w - \frac{\eta}{2}v_2(w)), \; v_4(w) = v(w - \eta v_3(w)), \tag{29}$$

*is*

$$\dot{w} = -v(w) + O(\eta^4). \tag{30}$$

*Proof.* We use the following Taylor expansion:

$$v(w + \delta) = v(w) = \nabla v(w)\delta + \frac{1}{2}\nabla^2 v(w)(\delta, \delta) + \frac{1}{6}\nabla^3 v(w)(\delta, \delta, \delta) + O(\|\delta^4\|), \tag{31}$$

where $\nabla^2 v(w) : \mathbb{R}^d \times \mathbb{R}^d \to \mathbb{R}^d$ is a symmetric bilinear form, and $\nabla^3 v(w) : \mathbb{R}^d \times \mathbb{R}^d \times \mathbb{R}^d \to \mathbb{R}^d$ is a symmetric trilinear form. With the formula we have:

$$v_4(w) = v(w) - \eta\nabla v(w)v_3(w) + \frac{\eta^2}{2}\nabla^2 v(w)(v_3(w), v_3(w)) - \frac{\eta^3}{6}\nabla^3 v(w)(v_3(w), v_3(w), v_3(w)) + O(\eta^4), \tag{32}$$

$$v_3(w) = v(w) - \frac{\eta}{2}\nabla v(w)v_2(w) + \frac{\eta^2}{8}\nabla^2 v(w)(v_2(w), v_2(w)) - \frac{\eta^3}{48}\nabla^3 v(w)(v_2(w), v_2(w), v_2(w)) + O(\eta^4), \tag{33}$$

$$v_2(w) = v(w) - \frac{\eta}{2}\nabla v(w)v(w) + \frac{\eta^2}{8}\nabla^2 v(w)(v(w), v(w)) - \frac{\eta^3}{48}\nabla^3 v(w)(v(w), v(w), v(w)) + O(\eta^4). \tag{34}$$

Putting them together we have:

$$v_4(w) + 2v_3(w) + 2v_2(w) + v(w) = 6v(w) - \eta\nabla v(w)(v_3(w) + v_2(w) + v(w))$$
$$+ \frac{\eta^2}{2}\left(\nabla^2 v(w)(v_3(w), v_3(w)) + \frac{1}{2}\nabla^2 v(w)(v_2(w), v_2(w)) + \frac{1}{2}\nabla^2 v(w)(v(w), v(w))\right) +$$
$$- \frac{\eta^3}{4}\nabla^3 v(w)(v(w), v(w), v(w)) + O(\eta^4), \tag{35}$$

$$v_3(w) + v_2(w) + v(w) = 3v(w) - \frac{\eta}{2}\nabla v(w)(v_2(w) + v(w)) + \frac{\eta^2}{4}\nabla^2 v(w)(v(w), v(w)) + O(\eta^3), \tag{36}$$

$$v_2(w) + v(w) = 2v(w) - \frac{\eta}{2}\nabla v(w)v(w) + O(\eta^2). \tag{37}$$

Bringing Equation (37) into Equation (36), we obtain:

$$v_3(w) + v_2(w) + v(w) = 3v(w) - \eta\nabla v(w)v(w) + \frac{\eta^2}{4}\nabla^2 v(w)(v(w), v(w)) + \frac{\eta^2}{4}(\nabla v(w))^2 v(w) + O(\eta^3). \tag{38}$$

Putting Equation (38) and Equation (37) together, we have:

$$-\eta\nabla v(w)(v_3(w) + v_2(w) + v(w)) = -3\eta\nabla v(w)v(w) + \eta^2(\nabla v(w))^2 v(w)$$
$$- \frac{\eta^3}{4}\nabla v(w)\nabla^2 v(w)(v(w), v(w)) - \frac{\eta^3}{4}(\nabla v(w))^3 v(w) + O(\eta^4). \tag{39}$$

Moreover, we have

$$\nabla^2 v(w)(v_2(w), v_2(w)) = \nabla^2 v(w)(v - \frac{\eta}{2}\nabla v(w)v(w), v - \frac{\eta}{2}\nabla v(w)v(w)) + O(\eta^2)$$
$$= \nabla^2 v(w)(v(w), v(w)) - \eta\nabla^2 v(w)(\nabla v(w)v(w), v(w)) + O(\eta^2), \tag{40}$$

and similarly

$$\nabla^2 v(w)(v_3(w), v_3(w)) = \nabla^2 v(w)(v(w), v(w)) - \eta\nabla^2 v(w)(\nabla v(w)v(w), v(w)) + O(\eta^2). \tag{41}$$

Bringing Equation (39) and Equation (40) into Equation (35) results in

$$v_4(w) + 2v_3(w) + 2v_2(w) + v(w) = 6v(w) - 3\eta\nabla v(w)v(w) + \eta^2((\nabla v(w))^2 v(w) + \nabla^2 v(w)(v(w), v(w)))$$
$$- \frac{\eta^3}{4}((\nabla v(w))^3 v(w) + 3\nabla^2 v(w)(\nabla v(w)v(w), v(w))$$
$$+ \nabla v(w)\nabla^2 v(w)(v(w), v(w)) + \nabla^3 v(w)(v(w), v(w), v(w))) + O(\eta^4).$$
(42)

Let us now derive the high-resolution ODE. From Equation (28), we have:

$$\frac{w^+ - w}{\eta} = -\frac{1}{6}(v_4(w) + 2v_3(w) + 2v_2(w) + v(w)).$$
(43)

Let us assume that $w^+ = w(t + \eta)$ and $w = w(t)$. Expanding the left we have:

$$\dot{w} + \frac{\eta}{2}\ddot{w} + \frac{\eta^2}{6}\dddot{w} + \frac{\eta^3}{24}\ddddot{w}.$$
(44)

Let us assume that the high-resolution ODE up to $O(\eta^4)$ has the form:

$$\dot{w} = f_0(w) + \eta f_1(w) + \eta^2 f_2(w) + \eta^3 f_3(w) + O(\eta^4).$$
(45)

Taking derivatives on both sides, we have:

$$\ddot{w} = (\nabla f_0(w) + \eta\nabla f_1(w) + \eta^2\nabla f_2(w) + \eta^3\nabla f_3(w))\dot{w} + O(\eta^4)$$
$$= \nabla f_0(w)f_0(w) + \eta(\nabla f_1(w)f_0(w) + \nabla f_1(w)f_0(w)) + \eta^2(\nabla f_1(w)f_1(w) + \nabla f_0(w)f_2(w) + \nabla f_2(w)f_0(w))$$
$$+ O(\eta^3).$$
(46)

Comparing the order $O(1)$ on both sides of Equation (43) we have:

$$f_0(w) = -v(w), \ f_1(w) + \frac{1}{2}\nabla f_0(w)f_0(w) = \frac{1}{2}\nabla v(w)v(w),$$
(47)

which gives $f_0(w) = -v(w)$ and $f_1(w) = 0$. Bringing it back to Equation (46) we obtain:

$$\ddot{w} = \nabla v(w)v(w) + O(\eta^2).$$
(48)

We take the derivatives on both sides of Equation (48) to get:

$$\dddot{w} = \nabla(\nabla v(w)v(w))\dot{w} + O(\eta^2)$$
$$= -\nabla(\nabla v(w)v(w))v(w) + O(\eta^2).$$
(49)

Let us now compute $\nabla(\nabla v(w)v(w))v(w)$. We note that $\nabla(\nabla v(w)v(w))$ is a linear form and

$$\nabla v(w + \delta)v(w + \delta) = (\nabla v(w) + \nabla^2 v(w)\delta + o(\|\delta\|))(v(w) + \nabla v(w)\delta + o(\|\delta\|))$$
$$= \nabla v(w)v(w) + \nabla^2 v(w)(\delta, v(w)) + \nabla^2 v(w)\delta + o(\|\delta\|).$$
(50)

Therefore, we have

$$\dddot{w} = -\nabla(\nabla v(w)v(w))v(w) = -\nabla^2 v(w)(v(w), v(w)) - \nabla^2 v(w)v(w) + O(\eta^2).$$
(51)

With Equation (51) we can compare $O(\eta^2)$ on both sides on Equation (43) and obtain

$$f_2(w) = 0.$$
(52)

Finally, we take the derivative of Equation (51). Since

$$\nabla^2 v(w + \delta)(v(w + \delta), v(w + \delta)) = (\nabla^2 v(w) + \nabla^3 v(w)\delta)(v(w) + \nabla v(w)\delta, v(w) + \nabla v(w)\delta)$$
$$= \nabla^2 v(w)(v(w), v(w)) + \nabla^3 v(w)(\delta, v(w), v(w)) + 2\nabla^2 v(w)(v(w), \nabla v(w)\delta)$$
$$+ o(\|\delta\|),$$
(53)

we have:

$$\nabla(\nabla^2 v(w)(v(w), v(w)))v(w) = \nabla^3 v(w)(v(w), v(w), v(w)) + 2\nabla^2 v(w)(v(w), \nabla v(w)v(w)).$$
(54)

Similarly, since

$$(\nabla v(w + \delta))(\nabla v(w + \delta))v(w + \delta) = (\nabla v(w) + \nabla^2 v(w)\delta)(\nabla v(w) + \nabla^2 v(w)\delta)(v(w) + \nabla v(w)\delta)$$
$$= (\nabla v(w))^2 v(w) + \nabla^2 v(w)(\delta, \nabla v(w)v(w))$$
$$+ \nabla v(w)\nabla^2 v(w)(\delta, v(w)) + \nabla v(w))^3\delta + (\|\delta\|)$$
(55)

$$\nabla(\nabla v(w))^2 v(w)(v(w)) = \nabla^2 v(w)(v(w), \nabla v(w)v(w)) + \nabla v(w)\nabla^2 v(w)(v(w), v(w)) + \nabla v(w)^3 v(w). \quad (56)$$

Combining Equation (51), Equation (54) and Equation (56) gives:

$$\dddot{w} = \nabla v \ddot{w} \dot{w} = \nabla v(w)^3 v(w) + \nabla^3 v(w)(v(w), v(w), v(w)) + 3\nabla^2 v(w)(v(w), \nabla v(w)v(w))$$
$$+ \nabla v(w)\nabla^2 v(w)(v(w), v(w)). \quad (57)$$

Combining Equation (45), Equation (46), Equation (51) and Equation (57) and comparing the $O(\eta^3)$ term of Equation (43), we obtain

$$f_3(w) = 0. \quad (58)$$

Therefore, the high-resolution ODE is:

$$\dot{w} = -v(w) + O(\eta^4). \quad (59)$$

$\square$

## C  PROOFS AND ADDITIONAL THEORETICAL RESULTS

Several proofs in this section are based on the derivations of the high resolution ODEs that were proven in Appendix B.

**Theorem 4.** *Suppose $\phi : \mathbb{R}^n \to \mathbb{R}$ and $\phi$ is $C^2$. Suppose also any stationary point $x^*$ of $\phi$ s.t. $\nabla\phi(x^*) = 0$ is either:*

1. *a strict local minimum, i.e. $\nabla_{xx}^2 \phi(x^*) \succ 0$;*

2. *a strict saddle. (Definition 5)*

*Then, the gradient flow is attracted towards a local minimum of $\phi$.*

*Proof.*
$$\dot{x} = -\nabla\phi(x) \quad (60)$$

Define $g(x) := -\nabla\phi(x)$.

a) Let us first show that strict saddles are non-asymptotically stable fixed points. Thus, gradient descent is not attracted to them.

$$g(x) \approx -\nabla_{xx}^2\phi(x)(x - x^*) \qquad \text{since } \nabla\phi(x^*) = 0 \quad (61)$$
$$g(z) \approx -\Lambda(z - z^*) \qquad \text{where } z := U^T x \quad (62)$$

This implies $g(z)_i \approx \lambda_{\min}(\nabla_{xx}^2\phi(x^*))(z - z^*)_i > 0$. Thus, we just showed that strict saddles are non-asymptotically stable, as desired.

Let us now show that all the local minima are asymptotically stable, and gradient descent is then attracted to them.

b) Define $V(x) := \phi(x) - \phi(x^*)$ to be a Lyapunov function. We have $x^*$ is asymptotically stable if:

$$\nabla V(x)^T g(x) < 0 \quad \forall x \neq x^* \quad (63)$$

From which the result follows. (e.g $\nabla V(x)^T g(x) = -||g(x)||_2^2$).

Similar results with a different proof can be derived from Lee et al. (2016). $\square$

**Theorem 2.** *The high resolution ODE of GD with the GRL up to $O(\eta)$ is:*

$$\dot{w} = -v(w) - \frac{\eta}{2}\nabla v(w)v(w) \quad (64)$$

*Moreover, this is asymptotically stable (see Appendix A.1) at a stationary point $w^*$ (Proposition 3) iff for all eigenvalue written as $\lambda = a + ib \in Sp(-\nabla v(w^*))$, we have $0 > \eta(a^2 - b^2)/2 > a$.*

*Proof.* This theorem can be obtained by computing the Jacobian of the ODE dynamics and using the Hurwitz condition Lemma 1. Specifically, we have the Jacobian of the dynamics at the stationary point is:

$$-\nabla v(\omega^*) - \frac{\eta}{2}\nabla v(\omega^*)^2 \quad (65)$$

Take $\lambda = a + ib \in Sp(-\nabla v(\omega^*))$, we must have:

$$\Re[a + ib - \frac{\eta}{2}(-a - ib)^2] < 0, \text{where } \Re \text{ stands for the real part.} \tag{66}$$

then,

$$a - \frac{\eta}{2}(a^2 - b^2) < 0 \tag{67}$$

as desired. □

**Corollary 1.** *For the high resolution ODE of GD with GRL (i.e., Equation (64)) to be asymptotically stable, the learning rate $\eta$ should be upper bounded by:*

$$0 < \eta < \frac{-2a}{b^2 - a^2}, \tag{68}$$

*for all  $\lambda = a + ib \in Sp(-\nabla v(w^*))$ with large imaginary part (i.e. such that $|a| < |b|$).*

*Proof.* With $|a| < |b|$ where $b$ is the imaginary part of $\lambda \in \mathbb{C}$, an algebraic manipulation of Theorem 2. i.e. $\eta(b^2 - a^2) < -2a$ leads to the desired result. □

**Theorem 3.** *The high resolution ODE of any RK2 method up to $O(\eta)$, $\dot{w} = -v(w)$, is asymptotically stable if for all eigenvalues $\lambda = a + ib \in Sp(-\nabla v(w^*))$, we have $a < 0$.*

*Proof.* This theorem can be obtained by computing the Jacobian of the ODE dynamics (see Appendix B for the derivation) and using the Hurwitz condition Lemma 1. For details, see the proof of Theorem 2. □

Unlike the high resolution ODE of GD with GRL in Corollary 1, up to $O(\eta)$, there is no upper bound constraint on the learning rate. Therefore, at the cost of an additional extra-gradient computation, we are allowed to take a more aggressive step. We observe in practice that this leads to both faster convergence and better transfer performance.

**Lemma 3.** *Suppose the game $\mathcal{G}(\mathcal{I}, \Omega_i, J_i)$ is a potential game with potential function $\phi$. Suppose the minimizers of $\phi$ are either a local minimum or a strict saddle (see Definition 5). The gradient play dynamics converges to a local Nash Equilibrium.*

*Proof.* A potential game can be analyzed in terms of the implicit function $\phi(w)$ since $\nabla\phi(w) = v(w)$. From Theorem 4, we know the gradient flow in $\phi(w)$ converges to a point $w^*$ that is a local minimum. Thus, this follows from Proposition 6. □

**Proposition 4.** *The domain-adversarial game is neither a potential nor necessarily a purely adversarial game. Moreover, its gradient dynamics are not equivalent to the gradient flow.*

*Proof.* From appendix A.2, a sufficient and necessary condition for a game to be potential is $\nabla v(\omega)$ being symmetric. Computing $\nabla v(\omega)$ using Equation (4) leads to $\nabla v(\omega)$ being asymmetric, from which the first part of the result follows. Another way to see this, it is to notice a there is no potential function $\phi$ satisfying $\nabla\phi(\omega) = v(\omega)$ (because of the flip in the sign in $d_{s,t}$). Since the game is not a potential game then the vector field is not equal to the gradient flow (see also for more details Mazumdar et al. (2020). ) To see the game is not necessarily a purely adversarial game, it suffices to show an example for which this does not happen. See Appendix A.3 and example therein. □

## C.1    PROPOSED LEARNING ALGORITHM

---
**Algorithm 1 Pseudo-Code of the Proposed Learning Algorithm**

---
**Input:** source data $D_s$, target data $D_t$, player losses $J_i$, network weights $\{\omega_i\}_{i=1}^3$, learning rate $\eta$
**for** $t = 1$ **to** $T - 1$ **do**
    $x_s, y_s \sim D_s, x_t \sim D_t$
    $v = [\text{gradient}(x_s, x_t, y_s, J_i) \text{ for } i \text{ in } 1..3 ]$.
    ~~$\omega = \text{Gradient Descent}(v, w, \eta)$~~
    $\omega = \text{Runge-KuttaSolver}(v, w, \eta, \text{steps}=1)$
**end for**

---

## C.2 CO APPROXIMATES RK2 (HEUN'S METHOD)

We have the update rule of CO Mescheder et al. (2017):

$$\omega^+ = \omega - [\eta v(\omega) + \gamma \frac{1}{2} \nabla ||v(\omega)||_2^2] \tag{69}$$

where $\frac{1}{2} \nabla ||v(\omega)||_2^2 = \nabla v(w)^\top v(w)$

From RK2 (Heun's Method), we have:

$$w^+ = w - \frac{\eta}{2}(v(w) + v(w - \eta v(w))). \tag{70}$$

$$= w - \frac{\eta}{2}(2v(w) - \eta \nabla v(w)v(w) + O(\eta^2)) \tag{71}$$

If $-\nabla v(w) = \nabla v(w)^\top \implies \nabla v(w)$ skew-symmetric, we can see that Equation (71) approximates the CO optimization method from Mescheder et al. (2017). This assumption is however not necessarily true in our case which may explain why RK2 was better in experiments. Experimentally, for CO, the best transfer performance was obtained for $\gamma = 0.0001$. We also observed that CO is very sensitive to the choice of $\gamma$.

## D EXPERIMENTAL SETUP ADDITIONAL DETAILS

Our algorithm is implemented in Jax Bradbury et al. (2018) (Digits, NLP benchmark) and PyTorch (Visual Task). Experiments are conducted on a NVIDIA Titan V and V100 GPU Cards.

The digits benchmark constitutes of two digits datasets **MNIST** and **USPS** LeCun et al. (1998); Long et al. (2018) with two transfer tasks (M → U and U → M). We adopt the splits and evaluation protocol from Long et al. (2018) which uses 60,000 and 7,291 training images and the standard test set of 10,000 and 2,007 test images for MNIST and USPS, respectively. We follow the standard implementation details of Long et al. (2018) for this benchmark. Therefore, we use LeNet LeCun et al. (1998) as the backbone architecture, dropout (0.5), fix the batch size to 32, the number of iterations per epoch to 937 and use weight decay (0.005). For GD-NM, we use the default momentum value (0.9). We follow the same approach for the additional hyper-parameters of Adam.

**Grid Search Details for Comparison vs Standard Optimizers.** For the learning rate, learning schedule and adaptation coefficient ($\lambda$), we run the following grid search and select the best hyper-parameters for each optimizer.

- learning rate: $0.0001, 0.001, 0.01, 0.03, 0.1, 0.3, 0.4$.
- learning scheduler: none, polynomial.
- adaptation coefficient ($\lambda$): $0.1, 0.5, 1$.

The grid is run using Ray Tune Liaw et al. (2018) and on the transfer task M→U. To avoid expending resources in bad runs we use a HyperBandScheduler Li et al. (2018). The same hyper-parameters are used for the other task (i.e U → M). The best criteria correspond to the transfer accuracy (as it is typically done in UDA).

**Grid Search Details for Comparison vs Game Optimization Algorithms.** Similar to the previous experiment, for the comparison vs game optimization algorithms, we run the following grid search in order to select their best parameters for the comparison. In the case of CO, we additionally add $\gamma$ extra-hypermeter to the grid as we experimentally found this optimizer to be sensitive to this parameter.

- learning rate: $0.0001, 0.001, 0.006, 0.01, 0.03, 0.1, 0.3, 0.4$.
- learning scheduler: none, polynomial.
- Adaptation coefficient ($\lambda$): $0.1, 0.5, 1$.
- CO Only. Additional ($\gamma$): $10, 1.0, 0.1, 0.01, 0.001, 0.001, \underline{0.0001}$

We show the best results of this grid search in Table 6. For EG, we observe training diverges for learning rates greater than 0.006.

**Experiment on Visda 2017.** For the visual tasks our implementation is built on PyTorch(1.5.1) and on top of the public available implementations of Long et al. (2018); Zhang et al. (2019); Junguang Jiang (2020). We did not run grid search here as it would be very computationally expensive. For the optimizer parameters, we tune thoroughly GD with Nesterov Momentum following the recommendation and implementation from Long et al. (2018); Zhang et al. (2019). For our method, we keep the exact same hyper-parameters of GD with NM, but

increase the learning rate. We do not experiment with RK4 since the performance was similar to RK2 in the other benchmarks (see Tables 3 and 4), and it is more computationally expensive.

**Experiment on Visda 2017 with SoTA algorithms.** For the comparison using SoTA algorithms, we use the well-tuned hyper-parameters from Jiang et al. (2020) and set the discrepancy measure to be $f$-DAL Pearson as in Acuna et al. (2021) since this empirically was shown to be better than both JS and $\gamma$-JS/MDD. We use Implicit Alignment Jiang et al. (2020) as this method also allows to account for the dissimilarity between the labels marginals. We did not run grid search here as it would be very computationally expensive. In our case, we only increase the learning rate to 0.2.

**Experiment on Natural Language.** In this experiment, we use the Amazon product reviews dataset Blitzer et al. (2006) that contains online reviews of products collected on the Amazon website. We follow the splits and evaluation protocol from Courty et al. (2017); Dhouib et al. (2020) and choose 4 of its subsets corresponding to different product categories, namely: books (B), dvd (D), electronics (E) and kitchen (K). As in Courty et al. (2017); Dhouib et al. (2020), from where we took the baseline result, the network architecture is a two layer MLP with sigmoid function. In our case, hyper-parameters are kept the same but the learning rate is increased by a factor of 10.

**Implementation on different frameworks.** We use both Jax and Pytorch in our paper for simplicity. Since the proposed method only requires changing some lines of code, it can be easily implemented in different frameworks. Prototyping in Jax small scale experiments such as the one in the Digits datasets is very easy. We then use PyTorch in order to integrate our optimizers with large-scale SoTA UDA methods (their codebase was implemented in PyTorch). This shows the versatility and simplicity of implementation of high-order ODE solvers. We did not see any issue while working with either Jax or Pytorch. Our choice of framework was simply based on modifying existing source code to evaluate our optimizer.

**Remark on error bars.** For experimental analysis involving a grid-search, we use the same seed and identically initialize the networks. That is, we give every optimizer the same fair opportunity to win as reported above. Note that reporting avg and std from a grid-search significantly increases the computational demand so we did not do this. That said, we report avg and std for our main experiments using the best parameters found on the grid-search.

# E  ADDITIONAL EXPERIMENTS

## E.1  NATURAL LANGUAGE PROCESSING TASKS.

We also evaluate our approach on natural language processing tasks. We use the Amazon product reviews dataset (Blitzer et al., 2006). We follow the splits and evaluation protocol from Courty et al. (2017); Dhouib et al. (2020) and choose 4 of its subsets corresponding to different product categories, namely: books (B), dvd (D), electronics (E) and kitchen (K). In our case, hyper-parameters are the same provided by authors but the learning rate is increased by a factor of 10. Table 4 shows noticeable gains by replacing the optimizer with either RK2 or RK4.

| | B→D | B→E | B→K | D→B | D→E | D→K | E→B | E→D | E→K | K→B | K→D | K→E | Avg |
|---|---|---|---|---|---|---|---|---|---|---|---|---|---|
| DANN (Dhouib et al., 2020) | 80.6 | 74.7 | 76.7 | 74.7 | 73.8 | 76.5 | 71.8 | 72.6 | 85 | 71.8 | 73 | 84.7 | 76.3 |
| with RK2 | **82.3** | 74.3 | 75.7 | **80.2** | **79.2** | **79.6** | 72.3 | 74.2 | 86.5 | 72.6 | 73.6 | **86.2** | 78.1 |
| with RK4 | 81.6 | **75.0** | **78.1** | 79.4 | 78.3 | 80.0 | **74.1** | **74.6** | **86.7** | **73.5** | **74.6** | 86.0 | **78.5** |

Table 4: Accuracy (%) on the Amazon Reviews Sentiment Analysis Dataset (NLP)

## E.2  SENSITIVITY TO SAMPLING NOISE

Table 5: Sensitivity to Sampling Noise controlled by the batch size in the Visda Dataset. Resnet-50

| | Batch Size | Avg | Std |
|---|---|---|---|
| GD | 64 | 67.82 | 0.29 |
| | 128 | 67.50 | 0.11 |
| | 160 | 67.42 | 0.24 |
| Ours(RK2) | 64 | 73.20 | 0.36 |
| | 128 | 73.81 | 0.26 |
| | 160 | **74.18** | 0.15 |

Table 6: Comparison vs Game Optimization Algorithms (best result from the grid search)

| Algorithm | M→ U |
|---|---|
| GD | 90.0 |
| GD-NM | 93.2 |
| EG Korpelevich (1976) | 86.7 |
| CO Mescheder et al. (2017) | 93.8 |
| **Ours(RK2)** | 95.3 |
| **Ours(RK4)** | **95.9** |

### E.3 ADDITIONAL COMPARISON VS GAME OPTIMIZATION ALGORITHMS

In Table 5, we show the sensitivity of our method to sampling Noise controlled by the mini-batch size. Specifically, we show results for 64, 128, 160 ( with 160 being the bigger size we could fit in GPU memory). We also add results for GD. We observed that our method performs slightly better when the batch size was bigger.

Table 6 shows the results of the grid-search for the comparison vs game optimization algorithms. For each optimizer, these results correspond to their best (in terms of transfer performance) determined from the grid-search explained in Appendix E. We also add GD for comparison. Our method notably outperforms the baselines.

### E.4 CO VS GRADIENT DESCENT AND EXTRA-GRADIENT ALGORITHMS

In Table 6 we can observe that RK solvers outperform the baselines. Interestingly, we can also observe that CO outperforms both EG and GD-NM. In this section, we provide some intuition about why this may be the case in practice and conduct an additional experimental on this direction.

In the analysis presented in Section 4, we show that a better discretization implies better convergence to a local optimality (assuming $\omega^*$ exists, and that it is a strict local NE). Specifically, under those assumptions, we show that in both GD and EG methods we should put an upper bound to their learning rate (see Section 4 and appendix B.2). In Appendix C.2, on the other hand, we show CO can be interpreted in the limit as an approximation of the RK2 solver. Based on this, we believe in practice CO may approximate the continuous dynamics better than GD and EG. Particularly, if we have an additional hyperparameter ($\gamma$) to tune thoroughly. We believe this might also be the reason of why we found CO to be very sensitive to the $\gamma$ parameter. In Table 7, we compare the performance before and after removing the best performing $\gamma$ from the grid search. We can see for values others than the best $\gamma$, CO does not outperform GD-NM either.

Table 7: Performance of CO vs others. CO($\gamma$ =1e-3) corresponds to the best result after removing 1e-4 from the grid search.

| Method | M →U |
|---|---|
| GD | 90.0 |
| GD-NM | 93.2 |
| CO ($\gamma = $ 1e-4) | 93.8 |
| CO ($\gamma = $ 1e-3) | 91.6 |
| RK2 | **95.3** |

### E.5 WALL-CLOCK COMPARISON

Table 8: Average Time Per Iteration Comparison (Wall-Clock Comparison)

| Algorithm | Avg time per iteration |
|---|---|
| GD-NM | $0.26 \pm 0.003$ |
| Ours(RK2) | $0.49 \pm 0.008$ |

In Table 8, we show experimental results of a wall-clock comparison between RK2 and GD-NM. Specifically, we compute the average over 100 runs on a NVIDIA GPU: TITAN V, CudaVersion 10.1 and PyTorch version: 1.5.1. As we can see, the average increase in time is less than 2x slower in wall-clock time.This is in line with the claims made in our submission. We additionally emphasize that this time can be improved with more efficient implementations (e.g. Cuda). Moreover, many more efficient high-order ODE solvers exist and could be used in the future. Our work also inspires future research in this direction.

## F  PYTORCH PSEUDOCODE OF RK2 SOLVER

Below, we show how to implement the RK2 solver in PyTorch by simply modifying the existing implementation of SGD.

```python
# Based on https://github.com/pytorch/pytorch/blob/release/1.5/torch/optim/sgd.py
class RK2(torch.optim.optimizer):
  # ......
  def update_step(self,fwd_bwd_fn):
      """
      fwd_bwd_fn: fn that computes y=model(x); loss(y,yhat).backward()
      example:
        # training loop
        x, yhat=sample_from_dataloader()
        optimizer.zero_grad()

        def fwd_bwd_fn():
          y=model(x)
          loss=compute_loss(y,yhat)
          loss.backward()
          return loss,y

        fwd_bwd_fn()
        optimizer.update(fwd_bwd_fn)
      """
      #  We want to compute the update equation:
      #   w - n/2* v(w) - n/2 * v(w-n*v(w))
      temp_step = {}
      with torch.no_grad():
          for jj_, group in enumerate(self.param_groups):
              #ignore weight-decay for simplicity.

              for ii_, p in enumerate(group['params']):
                  if p.grad is None:
                      continue
                  d_p = p.grad   # v(w)

                  # first part of the step.
                  # w - n/2*v(w)
                  p1 = p.data.clone().detach()
                  p1.add_(d_p, alpha=-0.5 * group['lr'])

                  #storing for later used.
                  temp_step[f"{jj_}_{ii_}"] = p1

                  # Computing now w-n*v(w) for the second part of the step
                  p.add_(d_p, alpha=-1.0 * group['lr'])

                  # set to none
                  p.grad = None   # self.zero_grad()

      # extra step evaluation.
      # this function does model forward,compute loss and loss backward.
      loss, _ = fwd_bwd_fn()
      # gradients are now v(w-n*v(w))

      with torch.no_grad():
```

```python
for jj_, group in enumerate(self.param_groups):
    for ii_, p in enumerate(group['params']):
        if p.grad is None:
            continue
        d_p = p.grad # v(w-n*v(w))

        # unpacking the first part of the step.
        p1 = temp_step[f"{jj_}_{ii_}"]

        # updating p with the stored value from before.
        # this is equivalent to p = p1 = w-n/2 * v(w) (but sligthly faster)
        p.zero_()
        p.add_(p1)

        #w= w - n/2*v(w) - n/2 * v(w-n*v(w))
        p.add_(d_p, alpha=-0.5 * group['lr'])
```

