# OpenReview forum: "Domain Adversarial Training: A Game Perspective"
_ICLR.cc/2022/Conference — ICLR 2022 Poster_

### Official Review · Reviewer_1YN1 · 2021-10-30

**Correctness:** 4
**Technical Novelty And Significance:** 3
**Empirical Novelty And Significance:** 3
**Recommendation:** 8
**Confidence:** 3

**Main Review:**

Strengths:

* Well written and understandable
* Theoretical sound in both problem description and solution
* Analyzing domain adaptation learning from a game perspective and analyzing the stability of the optimizer with the gradient reversal layer model is novel
* Relevant related work is considered and compared
* The experimental part is sufficient and supports the theoretical claims

Weaknesses:
* The theoretical analysis is limited to full batch training while in practice as well in the experiments mini-batches are used. However, the authors mentioned it in the paper.
* As with gradient-based methods, this method could also converge to a non-Nash equilibrium as outlined in [1], however this is likely a rare case and not unique to this proposal

[1] Mazumdar et al. On Finding Local Nash Equilibria (and Only Local Nash Equilibria) in Zero-Sum Games. https://arxiv.org/abs/1901.00838


**Summary Of The Paper:**

This manuscript considers the adversarial domain adaptation training problem, specifically the gradient reversal method, from the perspective of game theory. The authors show that gradient-based optimizers without an upper bound on the learning rate violate asymptotic convergence guarantees to local NEs. The authors further show that these constraints can be lifted by higher order ODE solvers. In the experimental part, the authors evaluate their method i.e. Runge-Kutta ODE solvers of order 2 and 4 with different general and also game optimized gradient-based optimizers on a MNIST/USPS digits dataset. Furthermore, they show hyperparameter robustness of their method and finally, the method is tested on more complex image and NLP datasets and compared to current SOTA methods. Overall better results are achieved.


**Summary Of The Review:**

The proposed method is theoretical founded and the experimental part supports the claims and the analysis is novel. I vote for accept, good paper.

---

> ### Author Response · Authors · 2021-11-21
> **Thank you for the review and feedback**
>
> We thank the reviewer for the positive feedback, finding our paper “well-written and understandable”, “theoretical sound in both problem description and solution” and acknowledging the novelty of our analysis. We also appreciate the weaknesses highlighted. We agree, below we provide additional details.
>
> **Analysis on the full batch training setting.** We agree this is a limitation of our work. This is stated as a limitation in the manuscript.  As in Barrett & Dherin (2021); Lu (2020); Shi et al. (2018), our analysis is in the full-batch setting. We believe the derivation of the stochastic dynamics of stochastic discrete-time algorithms is significantly more complicated and it is beyond the scope of our work. That said,  because of this limitation, we additionally demonstrate with experiments that our results are also valid when there is stochasticity due to sampling noise in the mini-batch (see Figure 5, Table 5).
>
> **Limitation of gradient-based methods.** This is a good point. In the updated revision, we have added this brief discussion to the limitations section. We are also thankful for the great reference. We have cited it.

---

### Official Review · Reviewer_uzT5 · 2021-11-02

**Correctness:** 3
**Technical Novelty And Significance:** 4
**Empirical Novelty And Significance:** 4
**Recommendation:** 6
**Confidence:** 5

**Main Review:**

1. The authors claim that "The gradient field of Equation (2) and the game's vector field (see Section 3.2) are equivalent, making
the original interpretation of DAL and our three-player formulation equivalent." Why does field equivalence induce formulation equivalence? What is the exact definition of "equivalence" here?

2. The layout of the paper needs to be improved. For example, Example  2 should be an independent paragraph in the paper but not a "window" embedded in the other section. Some figures and tables have the same issue.


**Summary Of The Paper:**

This paper analyzes adversarial domain learning (DAL) from a game-theoretical perspective, where the optimal condition is defined as obtaining the local Nash equilibrium. From this view, the authors show that the standard optimization method in DAL can violate the asymptotic guarantees of the gradient-play dynamics, thus requiring careful tuning and small learning rates. Based on these analyses, this paper proposed to replace the existing optimization method with higher-order ordinary differential equation solvers. Both theoretical and experimental results show that the latter ODE method is more stable and allows for higher learning rates, leading to noticeable improvements in transfer performance and the number of training iterations.

**Summary Of The Review:**

This paper is well-written and easy to follow. The theoretical contribution is solid, and the experimental studies show the superiority of the proposed method on the domain transfer task. As a consequence, I think the quality of this paper is marginally above the acceptance threshold.

---

> ### Author Response · Authors · 2021-11-21
> **Thank you for the review and feedback**
>
> We thank the reviewer for the positive feedback and finding our paper “well-written and easy to follow”, and acknowledging that both technical and empirical contributions are significant and do not exist in prior works. We answer your questions below.
>
>
> **1. Equivalence between interpretations due to the same vector field:** The gradient with respect to the parameters of the original DAL objective (eq. 2) is identical to v(w) (eq. 5) *due to the Gradient Reversal Layer (GRL)*. Therefore, in both cases the behavior of the update rule `w-\eta v(w)` (gradient descent) is identical. Assuming the same initial conditions, they will reach the same solution. By equivalent, we mean the same dynamics and that the intermediate and final solutions will be the same. Following your recommendation, we have clarified this in the manuscript after introducing eq. 5 (changes are marked in blue).
>
> **2. Layout of example 2 and some small tables**. We apologize for this, it was because of space constraints. We have made our best effort, given the space constraints, to improve the layout as suggested. In the updated version, example 2 is an independent paragraph, and the margin and the size of embedded tables have been increased. To do this, we have moved the experimental details of the natural-language processing task to the supplementary material (E.1).

---

### Official Review · Reviewer_tiyE · 2021-11-03

**Correctness:** 4
**Technical Novelty And Significance:** 3
**Empirical Novelty And Significance:** 3
**Recommendation:** 8
**Confidence:** 2

**Main Review:**

(I am not a specialist in domain adversarial learning. Therefore I am not entirely sure about the novelty and impact of the work in this field).

The paper is very well written. The introduction of both fields of domain adversarial learning (sec 2) and game theory (sec 3) is done properly, with an adequate study of the related work. Moreover, the authors did a good job convincing about the complexity of the three-players game setting.

From this perspective, using known tools from game theory / variationally inequalities / numerical analysis for ODE discretization, the authors show necessary and sufficient conditions for the existence of local Nash equilibrium (prop. 1 and 2), showed that a modified dynamic of the gradient flow (equation (8) ) ensure global convergence of the Euler discretization, and developed an algorithm (RK-2, generalization of extra-gradient, equation (9)).

Finally, they discussed several approaches with solid numerical experiments on various problems.

In terms of novelty, there are no "new" theoretical results, properly speaking. The novelty here is the derivation of the domain adversarial learning into a game, which is not straightforward but simplifies its analysis. Thanks to this new perspective, I believe this could open new doors and research direction in this field. For this reason, and because the paper is particularly well written, I recommend the paper to be accepted.


*** Post rebuttal

I have read the authors' answer and decided to keep my score.

**Summary Of The Paper:**

The authors exhibit a strong link between game theory and domain-adversarial training. They show the optimal point in the latter is a Nash equilibrium of a three players game. From this perspective, the authors show that standard approaches, like gradient descent, cannot work in this setting as the method is known to be divergent in such a case. Instead, they propose to use Runge-Kutta methods (for example) to discretize the ODE, which gives insights for novel algorithms with better convergence guarantees.

**Summary Of The Review:**

The authors presented a link between game theory and domain-adversarial training. In similar topic, such as GAN, such link opened new research perspectives and impacted positively the area. I believe this could be also the case here.

---

> ### Author Response · Authors · 2021-11-21
> **Thank you for the review and feedback**
>
> We thank the reviewer for the very positive and encouraging feedback and finding that our work could open new doors and research at the intersection of both fields. Below we provide a small explanation about the impact of our work within the domain adaptation community and more specifically the DAL field.
>
> **Wrt impact within the DAL field.**  To the best of our knowledge, to date, the derivation/interpretation of the domain adversarial learning (DAL) as a game has not been done, nor a proper definition of what algorithmic optimality is in DAL has been provided. Theoretical and empirical analysis of why DAL is difficult to train has not been provided either. Our work fills this gap in the DAL literature. Last but not least, using the great tools the game-theoretical community has developed, our analysis reveals that by changing several lines of code (e.g. drop-in replacement of GD with high-order solvers), we could achieve up to 3.5% transfer performance improvement with less than half of the training iterations. This is on top of a SoTA method in a challenging dataset. This is also a very important and practical result.

---

### Official Review · Reviewer_LuBk · 2021-11-04

**Correctness:** 4
**Technical Novelty And Significance:** 2
**Empirical Novelty And Significance:** 3
**Recommendation:** 6
**Confidence:** 3

**Main Review:**

Strengths :
The game-theoretical formulation is natural for this important problem.

The experiments seem to support the asymptotic convergence guarantees of the optimizer.

The empirical results look somewhat significant.

The practical significance of this work reveals that using high-order solvers instead of Gradient Descent (GD) in this setting leads to better results.

Weakness:
The technical novelty is only marginally novel and quite straightforward.




**Summary Of The Paper:**

The setting in the paper is the classic unsupervised domain adaptation problem, where we are given a labeled sample from a source distribution and an unlabeled sample from a target distribution. The goal is to minimize the risk on the target distribution. Theoretical results led to a breakthrough in practice - the Domain Adversarial Learning architecture (Ganin et al., 2016).

The paper suggests looking at the paper from a game theory perspective. This is natural, as the objective is to minimize the loss on the source distribution while maximizing the distinction between the distributions.
The optimal solutions of the game are characterized by the local NE.

Motivated by the results from game theory, the authors suggest replacing Gradient Descent (due to its limitation in this optimization problem) with other optimizers - ODE (ordinary differential equation) solvers.

**Summary Of The Review:**

My vote is -  marginally above the acceptance threshold due to the aforementioned reasons.

---

> ### Author Response · Authors · 2021-11-21
> **Thanks for the review**
>
> We thank the reviewer for their constructive feedback. Below we provide further clarification with respect to the main concern.
>
> __Novelty of the contributions:__ We do not claim novelty for the game theoretical tools and game theoretical results used in the paper. We built on the tools and results this great community has developed.   Our novelty is on using those tools to better understand and propose a new perspective on domain adversarial learning. To the best of our knowledge, to date , the derivation/interpretation of the domain adversarial learning (DAL) as a game has not been done, even though it feels natural. A proper definition of what algorithmic optimality is in DAL has not been provided. Theoretical or empirical analysis of why optimization in DAL is difficult has not been provided either. Our work aims to fill this gap in the DAL literature by examining the problem through this novel perspective, and as pointed out by reviewer tiyE, this is not straightforward and simplifies the algorithmic analysis. In summary, quoting reviewer 1YN1 “analyzing domain adaptation learning from a game perspective and analyzing the stability of the optimizer with the gradient reversal layer model is novel”. Moreover, we believe that thanks to this new perspective on DAL, our work could open new doors and research at intersections of these two fields as it happened for GANs (also pointed out by reviewer tiyE).
>
> Last but not least, we believe that the practical significance of our work might have been overseen. Our work reveals that by using high-order solvers instead of Gradient Descent (GD) in DAL, we could achieve up to 3.5% improvement with less than half of the training iterations on top of a SoTA algorithm in a challenging non-toy dataset. In practice, replacing the optimizer is simple and requires modifying only several lines of code.

---

### Author Response · Authors · 2021-11-21
**General Response**

We thank the reviewers for their thoughtful comments on our work. We are very grateful for the comments provided. e.g. __(R-LuBK)__  *“the game-theoretical formulation is natural for this important problem” , __(R-tiyE)__ “thanks to this new perspective, I believe this could open new doors and research direction in this field.”, ”the paper is very well written.” __(R-uzT5)__ “this paper is well-written and easy to follow. The theoretical contribution is solid, and the experimental studies show the superiority of the proposed method on the domain transfer task.” __(R-1YN1)__ “well written and understandable”, “theoretical sound in both problem description and solution“, “analyzing domain adaptation learning from a game perspective and analyzing the stability of the optimizer with the gradient reversal layer model is novel“ . We also thank __(R-uzT5)__ for appreciating our contribution both technical and empirical and acknowledging the non-existence of this in prior work in the DAL literature.*

We also very much appreciate the provided constructive feedback. This has allowed us to make improvements to our paper. Specifically, we have uploaded a revised version incorporating the minor changes recommended by *__(R-uzT5)__ (wrt exact definition of equivalence and layout of example 2 ), and __(R-1YN1)__ (wrt missing reference/discussion). Following*  *__(R-uzT5)__,* in the updated version, example 2 is an independent paragraph, and the size of some embedded tables has been increased. As a result, we have moved the experimental details of the natural-language processing task to the supplementary material (E.1).  *Changes are marked in blue color.*

---

### Decision · Program_Chairs · 2022-01-20

**Decision:**

Accept (Poster)

**Comment:**

Summary (from reviewer uzT5): This paper analyzes adversarial domain learning (DAL) from a game-theoretical perspective, where the optimal condition is defined as obtaining the local Nash equilibrium. From this view, the authors show that the standard optimization method in DAL can violate the asymptotic guarantees of the gradient-play dynamics, thus requiring careful tuning and small learning rates. Based on these analyses, this paper proposed to replace the existing optimization method with higher-order ordinary differential equation solvers. Both theoretical and experimental results show that the latter ODE method is more stable and allows for higher learning rates, leading to noticeable improvements in transfer performance and the number of training iterations.

All reviewers appreciated the contributions of this paper and recommended acceptance. While the methods themselves are not novel, the game perspective applied to this problem appears to be and the use of higher-order solves yield interesting theoretical and empirical improvements.

== Additional comments ==

1) For the comparison vs. game optimization algorithms (Figure 3), it would be nice to normalize the x-axis so that one "epoch" yields comparable computational cost among the different methods (as RK4 and RK2 is much more expensive than EG or GD per mini-batch). Given that EG had such bad performance there, it would not change the conclusions; but the current scaling is still quite misleading. Same comments for Figure 2.

2) Note that modern approaches for stochastic extragradient recommend to use different step-sizes for the extrapolation step and the update step (see e.g. Hsieh et al. NeurIPS 2020 "Explore Aggressively, Update Conservatively: Stochastic Extragradient Methods with Variable Stepsize Scaling") I suspect that much bigger step-sizes could be used in this case while maintaining convergence, and this version should be added to Figure 3.

3) In "Related Work | Two-Player Zero-Sum Games" -> note that Gidel et al. 2019a provided all their convergence theory and methods for stochastic variational inequalities and thus it also applies to three-player games, unlike seems to be implied by this paragraph. In particular, all the algorithms they investigated (Extra-Adam amongst others) could also be applied to DAL. While I can see that the specifics of the objective in DAL might be different than for GAN optimization, it would be worthwhile to acknowledge these alternative approaches more clearly, and I encourage the DAL community to investigate their performance more exhaustively for DAL than what was done in this paper.